# Structures and immune recognition of Env trimers from two Asia prevalent HIV-1 CRFs

Jun Niu [1,2,3,7], Qi Wang[1,7], Wenwen Zhao[1,7], Bing Meng [1,7], Youwei Xu [1,7], Xianfang Zhang[1], Yi Feng[4], Qilian Qi[1], Yanling Hao[4], Xuan Zhang[1], Ying Liu[4], Jiangchao Xiang [1], Yiming Shao[4,5] ✉ & Bei Yang [1,6] ✉

Structure-guided immunofocusing HIV-1 vaccine design entails a comprehensive understanding of Envs from diverse HIV-1 subtypes, including circulating recombinant forms (CRFs). Here, we present the cryo-EM structures of Envs from two Asia prevalent CRFs (CRF01_AE and CRF07_BC) at 3.0 and 3.5 Å. We compare the structures and glycosylation patterns of Envs from different subtypes and perform cross-clade statistical analyses to reveal the unique features of CRF01_AE V1 region, which are associated with the resistance to certain bNAbs. We also solve a 4.1 Å cryo-EM structure of CRF01_AE Env in complex with F6, the first bNAb from CRF01_AE-infected individuals. F6 recognizes a gp120-gp41 spanning epitope to allosterically destabilize the Env trimer apex and weaken inter-protomer packing, which in turn hinders the receptor binding and induces Env trimer disassembly, demonstrating a dual mechanism of neutralization. These findings broaden our understanding of CRF Envs and shed lights on immunofocusing HIV-1 vaccine design.

Since 1980s, human immunodeficiency virus type 1 (HIV-1) has led to more than 80 million infections and around 40 million deaths worldwide (https://www.who.int/data/gho/data/themes/hiv-aids), ranking it one of the most devastating infectious agents in recent history. While the primary way out of the HIV-1 epidemic is an effective vaccine, the development of such preventives is marked by frequent setbacks, largely due to the inherently high mutation rates of HIV-1 genomes and the immune evasion capacity of the viral envelope glycoprotein (Env), which is the sole target of host humoral immune response and thus the focus of HIV-1 vaccine design[1,2].

Despite its essential role in mediating HIV-1 host entry, most antigenic surfaces on Env are either hypervariable or cloaked by an evolving glycan shield[2,3]. As such, natural infections or traditional vaccine approaches tend to induce humoral response with narrow breadth[1]. Nevertheless, the discoveries of broadly neutralizing antibodies (bNAbs) against HIV-1 reinvigorated the efforts in immunofocusing HIV-1 vaccine design[4,5]. bNAbs can broadly neutralize a wide variety of HIV strains despite the variations in their Env sequences, thus their epitopes represent conserved immune vulnerabilities on HIV-1 Envs[6]. In recent years, multiple atomic-level structures of Env trimers alone or with different bNAbs have been determined, revealing several conserved vulnerable sites on Env, such as the glycan-rich epitopes at V1/V2 apex and V3 base, CD4 binding site (CD4bs), gp120-gp41 interface, fusion peptide (FP), membrane-proximal external region (MPER) and the 'silent face' of Env[7,8]. Guided by these valuable structural definitions, Env proteins can be reversely engineered to specifically present the desired epitopes to immune system, thereby focusing the host immune response to those conserved vulnerable sites and eliciting effective and broad humoral response to protect against future acquisition of HIV-1 with diverse genetic backgrounds[9,10]. Notably, the discovery of new bNAbs and the delineation of their epitopes are continuously pursued, as these

[1]Shanghai Institute for Advanced Immunochemical Studies and School of Life Science and Technology, ShanghaiTech University, Shanghai 201210, China. [2]CAS Center for Excellence in Molecular Cell Science, Shanghai Institute of Biochemistry and Cell Biology, Chinese Academy of Sciences, Shanghai 200031, China. [3]University of Chinese Academy of Sciences, Beijing 100049, China. [4]State Key Laboratory for Infectious Disease Prevention and Control, National Center for AIDS/STD Control and Prevention, Chinese Center for Disease Control and Prevention, Beijing 102206, China. [5]Changping Laboratory, Beijing 102206, China. [6]Shanghai Clinical Research and Trial Center, Shanghai 201210, China. [7]These authors contributed equally: Jun Niu, Qi Wang, Wenwen Zhao, Bing Meng, Youwei Xu. ✉e-mail: shaoyiming@cpl.ac.cn; yangbei@shanghaitech.edu.cn

studies would offer new opportunities to apply the structure-based immunofocusing approach[11].

Moreover, through combining the structural definitions of bNAbs epitopes and the bNAbs neutralization data on different viruses, Env signatures can be associated with bNAbs sensitivities and incorporated into vaccine design to further enhance the breadth of induced antibody responses[12]. Notably, bNAbs also hold great potential to serve as prophylactic or therapeutic agents on their own[13]. Indeed, a recent study indicates that passive transfer of bNAbs combinations provided sustained virological suppression in individuals with HIV[14]. In this regard, the bNAb-sensitivity-associated Env signatures shall also help to predict the susceptibilities of regional circulating HIV-1 viruses to certain bNAbs or even inform on clinical trial design as a biomarker of preventive or therapeutic efficacy.

During its sustained global transmission in past decades, HIV-1 has evolved into various subtypes and their hybrids, i.e., circulating recombinant forms (CRFs)[15]. The global distribution of HIV-1 is heterogenous and manifests regional differences. For instance, subtype C dominates Southern Africa and India, and subtype B is prevalent in America and Western Europe[16]. In Southeast Asia and East Asia, most of the infections are caused by CRFs, especially CRF01_AE and CRF07_BC[16]. Currently, the majority of bNAbs research focuses on subtypes A, B and C and the structural delineation of the sole immunogen Env has also been heavily skewed towards these subtypes[7,17]. In contrast, very few bNAbs have been isolated from CRF-infected donors and atomic-level structures of CRFs Envs in their prefusion states are scarce. As the Env sequence diversity ranges between 20% and 35% among different subtypes and CRFs, traditional vaccines derived from individual isolates may not be sufficiently protective across different subtypes[18]. To this end, structure-guided bNAbs epitopes grafting onto a consensus Env framework may be a better way to promote the immunofocusing strategy in HIV-1 vaccine design, which nevertheless requires a better structural and antigenic understanding of Envs from other subtypes, including CRFs. Notably, the RV144 vaccine, designed upon components from CRF01_AE and subtype B viruses, remains the only clinical trial that produced statistically significant reduction in HIV infection rate, also encouraging further research into CRFs[19].

Here, we present the cryo-EM structures of Asia prevalent CRF01_AE and CRF07_BC Envs at 3.0 Å and 3.5 Å, respectively. The observed differences among these two Envs and the Envs of other subtypes then prompted us to perform cross-clade statistical analysis and find that the V1 loop in CRF01_AE is significantly longer and glycosylated more heavily than other subtypes. These two characters are associated with decreased sensitivities to certain bNAbs, suggesting that the regional distribution of HIV-1 subtypes should be considered in the epitopes-focusing vaccine design and the clinical use of bNAbs. We further present a 4.1 Å cryo-EM structure of CRF01_AE Env in complex with bNAb F6, which is the first bNAb obtained from CRF01_AE-infected individuals[20]. Comparing the epitope of F6 to those of other interface bNAbs reveals that the glycan free area near F6 epitope is a conserved site of vulnerability on Env and holds potential for immunofocusing vaccine design. We also reveal that F6 binding could allosterically hinder the binding of receptor CD4 and induce Env trimer disassembly, demonstrating a dual mechanism of neutralization. Collectively, these findings broaden our understanding of HIV-1 CRFs and shed lights on structure-based immunofocusing vaccine design.

## Results
### Cryo-EM structures of CRF01_AE and CRF07_BC Envs
The Env gene sequences representing subtypes CRF01_AE and CRF07_BC were from virus isolates BJOX018000.e12 (X18, GenBank ID: AIS42911.1) and BJOX016000.e15 (X16, GenBank ID: AIS42666.1), each obtained through single genome amplification (SGA) from acute/early HIV-1 infected individuals in China[21]. To overcome Env metastability

and create stable, homogeneous Env trimers for structural studies, uncleaved prefusion-optimized (UFO) design was applied to these two Env sequences[22], generating X18 UFO (CRF01_AE Env) and X16 UFO (CRF07_BC Env) (Supplementary Fig. 1a). We complexed X18 UFO and X16 UFO with bNAb 8ANC195 to further stabilize the Env trimers in prefusion state (Supplementary Fig. 1b, c) and determined their cryo-EM structures at 3.0 Å and 3.5 Å, respectively (Fig. 1a, b, Supplementary Fig. 1d–i and Supplementary Table 1). The overall structures of X18 UFO and X16 UFO adopt closed trimeric conformation in prefusion state, and closely resemble the Env structures of representative strains BG505 (subtype A), JRFL (subtype B), 16055 (subtype C) and X1193.c1 (subtype G) (Fig. 1c)[23–28]. Pair-wise structural alignments revealed low Cα root-mean-square deviation (RMSD) values for both protomers and trimers among these Envs (Supplementary Fig. 1j), further corroborating the overall structural homology of Envs across subtypes.

Despite the overall structural similarities, local conformational variations exist among X18 UFO, X16 UFO and the Envs of representative strains from other subtypes, most notably in the heptad repeat 1 N-terminal ($HR1_N$, 548-568) region, FP region and surface-exposed hypervariable loops V1, V2, V4 and V5 (Fig. 1d). While the structural differences in the hypervariable loops are consistent with their high sequence variations, the conformational differences in the $HR1_N$ region is largely attributable to the adoption of native-like trimer designs[29], wherein the engineering of proximal regions in SOSIP[30], NFL[31] or UFO[22] design all disrupt the native α-helical conformation of $HR1_N$ (Supplementary Fig. 1k, compare others to JRFL WT). The FPs are generally 'exposed and disordered' in the closed state of prefusion Env trimers but would rearrange to become 'buried' in the CD4-induced open state of Env[32]. Consistent with their closed prefusion conformations, the FPs in X18 UFO and X16 UFO are both 'exposed', akin to the FP conformation in native prefusion Env trimers (Supplementary Fig. 1l, compared to JRFL WT). Previously, a rare 'buried' conformation of FP has been captured in a 3.9 Å prefusion structure of a very early transmitted founder virus (CRF01_AE T/F100), which was thought to represent an intermediate between the closed and open states[33]. Here, the FP of X18 UFO (also from CRF01_AE) did not recapitulate the 'buried' intermediate state seen in T/F100 SOSIP trimer, possibly due that the UFO trimer is not cleaved between gp120 and gp41 as in SOSIP trimer.

We also compared the interfaces between 8ANC195 and X18 UFO (CRF01_AE), X16 UFO (CRF07_BC) or BG505 Env (subtype A), and found that amino acid (AA) variations of interface residues among the three Envs do not affect the binding of 8ANC195 much (Supplementary Fig. 2a–f). Moreover, we calculated the AA usage frequencies of CRF01_AE ($n = 618$) and CRF07_BC ($n = 56$) viruses in 8ANC195-contacting region and compared them to Env signatures associated with 8ANC195 sensitivity (Supplementary Fig. 2g), which indicates that the sensitivities of CRF01_AE and CRF07_BC viruses to 8ANC195 would not significantly differ from other M-group viruses.

### Glycosylation features of representative strains from different subtypes
The HIV-1 Envs are heavily glycosylated. For neutralization, bNAbs have to accommodate or even specifically recognize certain glycans within the glycan shield of Env[6,34]. We thus analyzed the N-linked glycans on X18 UFO and X16 UFO in detail. There are 29-30 potential N-linked glycosylation sites (PNGS) in each protomer of X18 or X16, we observed 18-20 of them in the cryo-EM structures (Fig. 1e) and confirmed the existence of other PNGS, most of which locate in or near the structurally invisible variable loops, through mass spectrometry analysis (see methods). Together, the structure and MS analysis identified 30 of the 30 PNGS in each protomer of X18 Env and 24 of the 29 PNGS in X16 Env (Fig. 1e). Of these confirmed PNGS, 16 are conserved (present in > 85% of HIV-1 Env sequences, $n = 6515$) among M-group viruses (Fig. 1e, yellow).

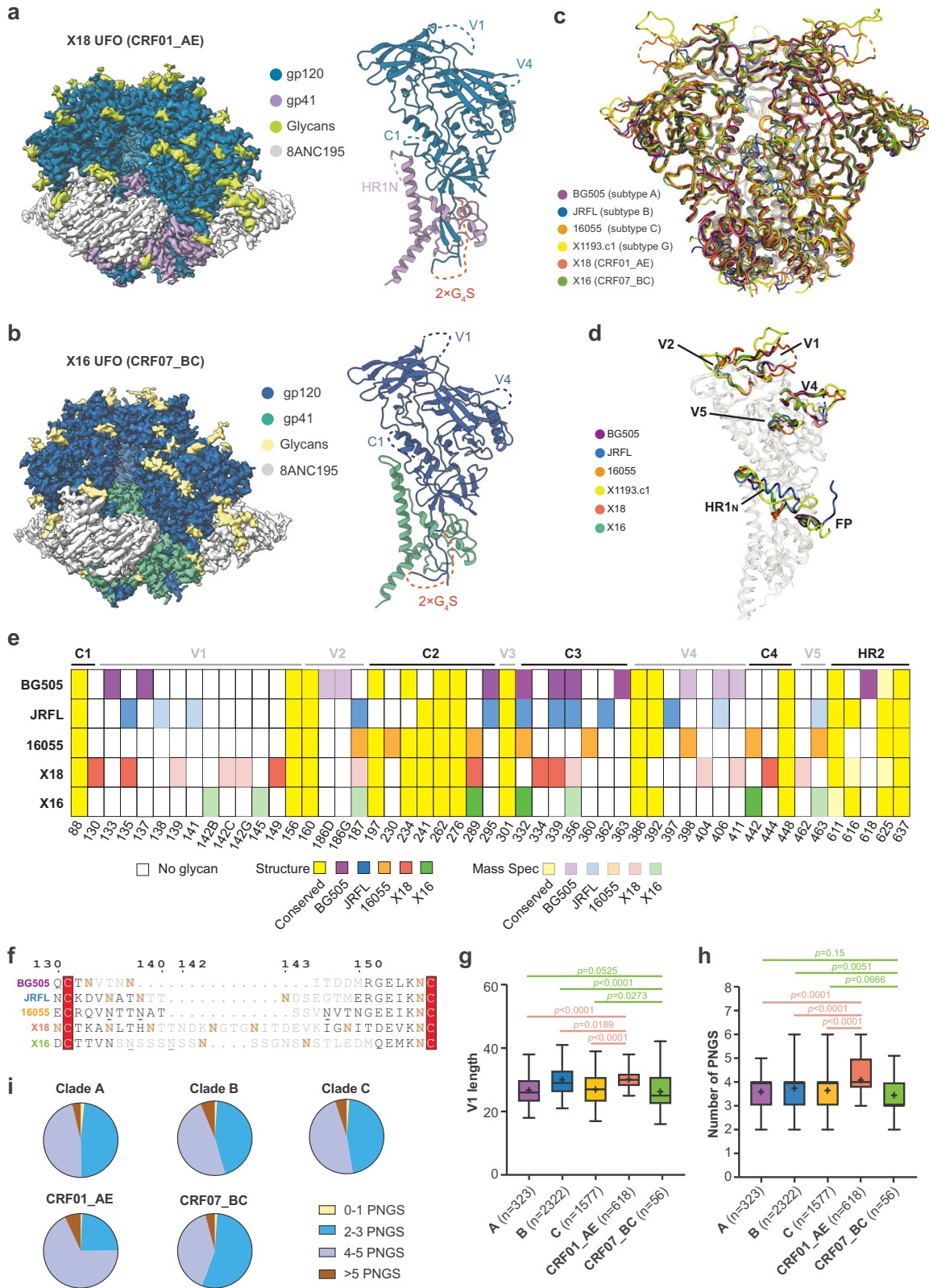

As previously reported, CRF01_AE and CRF07_BC are recombinants of subtypes A/E and subtypes B/C respectively[35,36]. We thus compared the glycan shields of X18 (CRF01_AE) and X16 (CRF07_BC) to those of representative strains from subtypes A (BG505), B (JRFL) and C (16055)[23,25,37,38] (Fig. 1e). While conserved glycosylation sites generally locate in the sequence-conserved C1, C2 and heptad repeat 2 (HR2) regions (Fig. 1e, yellow), most of the strain-specific N-linked glycans cluster around hypervariable loops, except for V3 wherein N301 glycosylation is highly conserved (Fig. 1e). Of note, the V1 of X18 (CRF01_AE) is glycosylated at 6 different sites, making it the most heavily glycosylated V1 among the five strains (Fig. 1e, compared to 2-3 such sites in other strains). The conserved C3 and C4 regions also manifested strain-specific glycosylation patterns, wherein local shifting of glycosylation sites are the most common manifestations

**Fig. 1 | Cryo-EM structures and cross-subtype analyses reveal the unique features of V1 in CRF01_AE Envs. a–b** Cryo-EM maps (side views) of X18 UFO (**a**) and X16 UFO (**b**) in complex with 8ANC195 Fab are shown on the left. The reconstituted models of X18 UFO (**a**) and X16 UFO (**b**) protomers are shown on the right, wherein areas invisible in structures are indicated with dashed lines and labeled. **c** The atomic models of X18 UFO (CRF01_AE) and X16 UFO (CRF07_BC) trimers are shown as ribbon diagrams and superimposed onto the atomic models of BG505 SOSIP[28] (subtype A, PDB:5CJX), JRFL SOSIP[27] (subtype B, PDB: 5FYK), 16055 NFL[23] (subtype C, PDB:5UM8) and X1193.c1 SOSIP[27] (subtype G, PDB:5FYJ) Env trimers. **d** Superimposition of X18 UFO, X16 UFO, BG505 SOSIP, JRFL natively cleaved[25] (PDB:5FUU), 16055 NFL and X1193.c1 SOSIP Env protomers highlights the conformational differences in surface-exposed variable loops, as well as FP and HR1$_N$ regions. For clarity, the superimposed models are rendered transparent except for these regions. **e** PNGS comparison among BG505, JRFL, 16055, X18 and X16. PNGS

that are visible in structures or confirmed by mass spectrometry are represented by different shades of indicated colors. Those conserved PNGS are colored yellow. Glycan positions are numbered according to HXB2 numbering. MS data is either from this study or from previous publications[37]. **f** Sequence alignment is shown for BG505, JRFL, 16055, X18 and X16 V1 loops, with residues invisible in structures colored grey. Confirmed PNGS are in bold brown, unconfirmed PNGS are underlined. **g–h** Box plots of the V1 loop length (**g**) and PNGS (**h**) of subtypes A (n = 323), B (n = 2322), C (n = 1577), CRF01_AE (n = 618) and CRF07_BC (n = 56). The box and whisker represent the 25–75% and 5–95% percentile respectively, the median is shown as bold line and the mean is depicted with '+'. P values are given by the Mann–Whitney test. **i** Pie plots depicting the fractions of V1 loops with 0–1, 2–3, 4–5 and >5 PNGS in each HIV-1 subtype. HIV-1 Env sequences are from the Los Alamos HIV sequence database (www.hiv.lanl.gov). Source data are provided as a Source Data file.

(Fig. 1e). For instance, while strain BG505 is glycosylated at N363, this glycosylation site is shifted to N362 and N360 in strain JRFL and strain 16055 respectively (Fig. 1e). In C4 region, N-glycan is also shifted from N442 in strain 16055 and X16 to N444 in X18 (Fig. 1e). Notably, local shifting of glycosylation sites may be a feasible strategy for HIV-1 Env to evade the recognition of neutralizing antibodies while still maintaining the integrity of glycan shield nearby[34].

### The unique features of CRF01_AE V1 region

As noted above, the V1 loop in X18 hosts more PNGS than other strains (Fig. 1e, f). Moreover, the V1s of X18 (CRF01_AE) and X16 (CRF07_BC) are significantly longer than those of strains BG505 (subtype A), JRFL (subtype B) and 16055 (subtype C) (Fig. 1f). Next, we sought to explore whether these strain-specific features of X18 and X16 V1s could be generalized to corresponding subtypes. We first compared the V1 lengths in CRF01_AE (n = 618) and CRF07_BC (n = 56) Envs to those in subtypes A (n = 323), B (n = 2322) and C (n = 1577) Envs (Fig. 1g). The data indicates that subtype CRF01_AE indeed tends to have a longer V1 than subtypes A, B and C (Fig. 1g). On the other hand, the V1 length in CRF07_BC does not deviate much from that in subtype A, although it seems to be shorter than that in subtypes B and C (Fig. 1g). We then compared the PNGS numbers in V1s of CRF01_AE and CRF07_BC Envs to those of subtype A, B and C Envs and found that the differences between CRF01_AE and subtypes A, B and C are all highly significant (Fig. 1h). The percentage of Envs hosting 4 or more V1 glycans reaches 75% in CRF01_AE while this number only ranges 50-54% in subtypes A, B and C (Fig. 1i). Meanwhile, the average V1 PNGS number in CRF07_BC is lower than that in subtype B, although the difference between CRF07_BC and subtype A or C is not statistically significant (Fig. 1h). Consistently, the percentage of Envs hosting 4 or more V1 glycans in CRF07_BC is lower (Fig. 1i, 44% in CRF07_BC versus 50-54% in subtypes A, B and C). Thus, the cross-subtype statistical analysis of M-group HIV-1 Env sequences revealed that the V1 loop of CRF01_AE is significantly longer and glycosylated heavier than the V1s of subtypes A, B and C.

### V1 features are associated with sensitivities to certain bNAbs

The Env molecular features near epitope can impact the function of corresponding bNAbs. For instance, extended length and higher number of PNGS within V5 have been correlated with increased resistance to CD4bs bNAbs, which bind near the V5 region[12]. Considering that the V1 region locates near the epitopes of V3, V1/V2 apex and CD4bs bNAbs (Fig. 2a), V1 features like length or glycosylation level may influence the binding of these bNAbs. To test this hypothesis, we selected 1–2 representative bNAbs from each group of V3, Apex, CD4bs, gp120-gp41interface and FP bNAbs and characterized their interactions with X18 (CRF01_AE), X16 (CRF07_BC) or Envs of subtypes A, B and C representative strains (Fig. 2b and Supplementary Fig. 3a). We found that the affinities between X18 and PGT121 (V3 bNAb), PG9 (Apex bNAb) or VRC01 (CD4bs bNAb) are all obviously lower than those between other Envs and these bNAbs (Fig. 2b). In contrast, the

interactions between X18 and interface bNAbs 8ANC195, 35O22 or FP bNAb ACS202 are all close to average as compared to other Envs (Fig. 2b). These results suggest that a long and heavily glycosylated V1 likely renders X18 less sensitive to the tested V3, Apex and CD4bs bNAbs.

To confirm the correlation between the V1 molecular features and the viral sensitivities to V3, Apex and CD4bs bNAbs, we then calculated the association between V1 length (or PNGS number) and measured neutralization potency of prototypical V3, Apex or CD4bs bNAbs on M-group HIV-1 viruses (Fig. 2c–e). We found that the V1 length correlates inversely with the viral sensitives to representative V3 bNAbs from three different lineages (Fig. 2c, top panels and Supplementary Fig. 3b). Also, more PNGS in V1 strongly correlates with decreased sensitivity to V3 bNAbs PGT128, 10.1074 and DH270 (Fig. 2c, bottom panels), though not to PGT121 (Supplementary Fig. 3b).

For prototypical Apex bNAbs PG9, VRC26.25 and PGDM1400, the V1 lengths are also inversely correlated with their neutralizing potency (Fig. 2d, top panels). Some weak correlations between the V1 PNGS numbers and resistance to these bNAbs can also be identified (Fig. 2d, bottom panels). We also found that increased V1 length is linked with decreased sensitivity to VRC01 and CH235.12, two representative CD4bs bNAbs (Fig. 2e, top panels), consistent with previous findings that long V1 and V2 regions can mediate in vivo escape to CD4bs bNAbs[39]. The correlation between PNGS numbers and CD4bs bNAbs sensitivities is bNAb specific, as a correlation was observed for VRC01 while no correlation was found for CH235.12 (Fig. 2e, bottom panels). No obvious correlation was identified between the molecular features of V1 and the neutralizing potency of representative interface or FP bNAbs (Fig. 2f, g), which is conceivable as the epitopes of theses bNAbs all locate far from V1.

The above results thus indicate a significantly higher chance for CRF01_AE viruses to be resistant to V3 and Apex bNAbs, given the unique molecular features of their V1s. These results also suggest that the regional distributions of subtypes should be considered in epitopes-focusing vaccine design and the prophylactic usage of corresponding bNAbs (discussed later in detail).

### bNAb F6 recognizes a gp120-gp41 interface epitope

Thus far, very few bNAbs are isolated from CRFs-infected donors, limiting our understanding of the host immune response against CRFs infections. To further extend our antigenic understanding of CRF Envs, we then solved the cryo-EM structure of X18 UFO (CRF01_AE) in complex with F6, which is the first bNAb (57% breadth) obtained from a CRF01_AE-infected individual[20]. After extensive cryogenic condition optimization, we were able to reconstitute a 4.14 Å cryo-EM map of X18 UFO-F6 complex wherein three F6 bind one X18 UFO trimer (Fig. 3a, Supplementary Fig. 4a–d and Supplementary Table 1). In the reconstituted cryo-EM map, the density of the Env apex proximal regions (V1/V2/V3 and part of C4, V4, V5) is shattered and untraceable, indicating high flexibility (Fig. 3a). In contrast, the densities of other Env

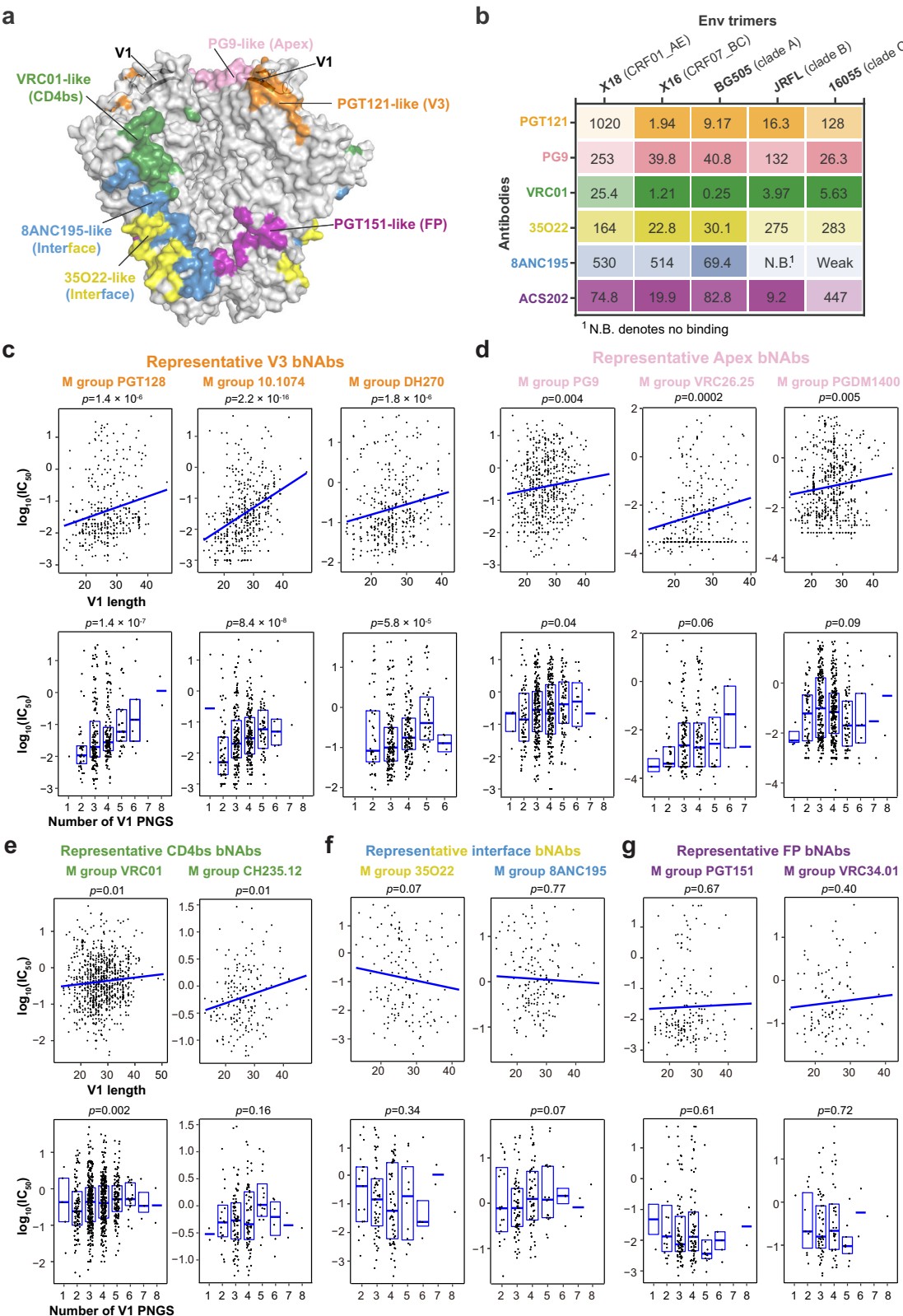

regions, as well as the variable domains of F6, are all sufficiently good for unambiguous model building, especially at the interface after local refinement (Supplementary Fig. 4e–g). To facilitate the model building process, we also solved a 3.3 Å crystal structure of bNAb F6 alone (Supplementary Table 2). This structure and the X18 UFO structure solved in this study were then docked into the EM map and further refined to generate the atomic model of the X18 UFO-F6 complex.

Akin to other bNAbs, the epitope of F6 also involves N-glycans. Specifically, the conserved N88 glycan packs against the heavy chain framework region 1 (HFR1) of F6 and contributes ~350 Å² buried epitope areas (Fig. 3b, close-up view 1). When N88 is mutated to Ala, the association rate between F6 and X18 UFO roughly doubles, while the disassociation rate also speeds up about twofold, leaving the $K_D$ largely unchanged (Fig. 3d and Supplementary Fig. 5a). Such phenomenon is

**Fig. 2 | V1 loop length and PNGS number are associated with the sensitivities to V3 and Apex bNAbs. a** Surface representation of prefusion Env trimer colored by the indicated bNAbs epitopes, with V1 loops in two neighboring Env protomers shown in cartoon. Note that the V1 loop locates in proximity to V3, Apex and CD4bs bNAbs epitopes and thus may affect the binding of these bNAbs. **b** The binding affinities between indicated bNAbs and Env trimers from representative strains of different subtypes are determined by SPR and shown as $K_D$ (nM). Lighter shades of color indicate lower affinity and vice versa. X18 UFO (CRF01_AE) is less sensitive to PGT121 (V3 bNAb), PG9 (Apex bNAb) and VRC01 (CD4bs bNAb) as compared to the Envs of representative strains from other subtypes. **c–g** Correlations between V1 loop characteristics and bNAb sensitivities are represented by scatter plots. Top:

Regression lines were fitted using the ggplot2 package in R (stat_smooth function). Bottom: The box represent the 25–75% percentile and the median is shown as bold line. HIV-1 viruses neutralization data and V1 loop characteristics of corresponding viruses for representative V3 bNAbs (**c**), Apex bNAbs (**d**), CD4bs bNAbs (**e**), interface bNAbs (**f**) and FP bNAbs (**g**) are prepared as described in Methods and stored in Source Data. The *p* values are calculated from Kendall's tau. The sample size are as following (n represents biologically independent samples): PGT128 (*n* = 348), 10.1074 (*n* = 339), DH270 (*n* = 366), PG9 (*n* = 546), VRC26.25 (*n* = 288), PGDM1400 (*n* = 570), VRC01 (*n* = 840), CH235.12 (*n* = 186), 35O22 (*n* = 148), 8ANC195 (*n* = 166), PGT151 (*n* = 189) and VRC34.01 (*n* = 110). Source data are provided as a Source Data file.

---

reminiscent of 3BC315, a gp41-binding bNAb isolated from clade B infected individuals, which also shows synchronously increased on- and off-rates in the absence of glycan[40]. Hence, as previously proposed, although N88 glycan is not required for F6 and 3BC315 to bind and may even sterically restrict their access, it may work as a 'clasp' to help keep antibodies in place once they are bound[40].

F6 also utilizes its complementarity-determining region (CDR) 1 and 2 in light chain (CDRL1 and CDRL2) to simultaneously engage the α8 helix of gp41 and the C-terminus of gp120 (Fig. 3b, close-up view 2). Of all the Env residues at this interface, R500 from gp120 plays a central role. The aliphatic part of R500 and the aromatic ring of Y619 from α8 helix form a hydrophobic cluster with the sidechains of Y49 and L50 from CDRL2, zipping α8 (gp41), the C-terminus of gp120 and CDRL2 together. Meanwhile, R500 also forms electrostatic interactions with E53 from CDRL2 and D30 from CDRL1, further strengthening the adhesion of F6. When R500 was mutated into Ala, a 90% affinity decrease was observed, corroborating its importance (Fig. 3d and Supplementary Fig. 5a). Compared to the interface between the Env and the light chain CDRs, the interface between the Env and the heavy chain CDRs (CDRHs) is even larger, and the interaction is mainly driven by hydrophobic packings (Fig. 3b, close-up view 3). The CDRH3 adopts an anti-parallel β-sheet conformation and side-wedges into a hydrophobic cleft formed by residues I535, I603, Y619 and W623 from α6-α8 helices of gp41 and residues Y39 and T499 from β4̄/β26 sheets of gp120 (Fig. 3b, close-up view 3). Besides CDRH3, Y32 from CDRH1 also forms hydrophobic interactions with the side chain of A532 in the α6 helix (gp41) (Fig. 3b, close-up view 3). These structural analyses clearly show that F6 simultaneously engage residues and glycans from both gp120 and gp41, thereby categorizing it into the gp120-gp41 interface bNAbs (Fig. 3c). Indeed, Ala mutations on gp120 residues like Y39, T499 and R500 generally lead to 90% to 97% affinity decrease, and Ala mutations on gp41 residues like Y619, W623 and N624 also result in 82% to 99% affinity decrease, further validating the gp120-gp41 spanning epitope of F6 (Fig. 3d and Supplementary Fig. 5a).

### Env signatures associated with viral sensitivity to bNAb F6

To provide further information for immunofocusing vaccine design and potential clinical usage of F6, we then tried to retrieve the Env signatures associated with F6 sensitivity. Using the available F6 neutralization dataset[20], we calculated the correlations between viral molecular signatures near F6 epitope (*e.g.*, AAs usage and glycan signatures) and the sensitivities of corresponding viruses to F6. We found that K at residue 500 in gp120 is associated with F6 sensitivity while E at this position is linked to F6 resistance (Fig. 3e). Given that K and R are chemically similar, we postulated that R at this position shall also be associated with F6 sensitivity although the current correlation is not statistically significant, possibly due to the small size of present neutralization dataset. Indeed, wildtype X18, which bears an R at residue 500, binds F6 with high affinity (Fig. 3g, $K_D$ at 41 pM). In contrast, wild type X16, which bears an E at residue 500, does not bind F6 at all (Fig. 3g and Supplementary Fig. 5a). As shown above, R500 on Env forms salt bridges with D30 and E53 from F6 (Fig. 3b). Hence, the change of R500 to E500 would lead to electrostatic repulsion among

E500, D30 and E53 (Fig. 3f), thereby explaining the failure of X16 to bind F6 (Fig. 3g). To further support the revealed F6 sensitivity signatures, a R500E mutation almost eliminates the binding of F6 to X18 (0.7% of original affinity), while a E500R mutation endows X16 responsiveness to F6 (Fig. 3g and Supplementary Fig. 5a).

Next, we calculated the AA frequencies at residue 500 in subtype A, B, C, CRF01_AE and CRF07_BC Envs to predict the sensitivities of different HIV-1 subtypes to F6. The higher frequency of F6 resistant signature at residue 500 (i.e., E500) in CRF07_BC and subtype C viruses suggests that viruses from these two subtypes are more likely to be F6 resistant as compared to viruses from CRF01_AE, subtypes A and B (Fig. 3h).

### Comparison of F6 to other gp120-gp41 interface bNAbs

Next, we compared F6 to 35O22, 8ANC195, PG151 and 3BC315, bNAbs that have been shown to bind near the interface between gp120 and gp41[25,28,38,40]. The epitope of F6 is different from that defined by 8ANC195 and 35O22, two prototypical gp120-gp41 interface bNAbs, and is also distinct from the epitope of PGT151, a representative FP bNAb (Fig. 4a). Meanwhile, the binding position of F6 on Env is similar to that of 3BC315, although their approaching angles and orientations of heavy and light chains are different (Fig. 4a). We also compared F6 to 1C2, a cross-reacting antibody (87% breadth) elicited by heterologous Env trimer-liposome prime:boosting in rabbits[41] (Fig. 4a). The epitope of 1C2 overlaps that of F6 by around 953 Å$^2$ (Supplementary Fig. 5b) and their approaching angles are very similar (Fig. 4a). Nevertheless, although F6, 3BC315 and 1C2 recognize overlapping areas on Env surface, differences are obvious among them. In the context of Env trimer, 3BC315 and 1C2 interact with gp41 from two neighboring protomers simultaneously, while F6 only interact with gp41 from one protomer (Fig. 4b). Moreover, F6 makes extensive and key interactions with gp120 (Fig. 3b–g), while 1C2 only make limited interactions with gp120 and 3BC315 barely interacts with gp120 residues[40] (Fig. 4c). The different participation of gp120 also explains the differences in their neutralization mechanism (see discussion later).

The contributions of N-linked glycans to epitopes are also different among these antibodies (Fig. 4d). In the context of Env protomer, F6 buries 309 Å$^2$ and 748 Å$^2$ epitope areas on gp120 and gp41 respectively, while N-linked glycans only contribute 349 Å$^2$ to its buried epitope area and appear to be dispensable (Figs. 3d, 4d). In contrast, glycans are indispensable for the binding of 35O22, 8ANC195 and PGT151[42–44] and the contributions of glycans (1051 Å$^2$, 1657 Å$^2$ and 1878 Å$^2$) to 35O22, 8ANC195 and PGT151 epitopes all outweigh the contributions of protein part (748 Å$^2$, 1254 Å$^2$ and 1481 Å$^2$) (Fig. 4d). Akin to F6, the epitopes of 1C2 and 3BC315 only received limited contributions from glycans (Fig. 4d). Of note, the epitope of F6 locates near one of the two major functional gaps in the continuous glycan shield on Env surface. As the area near F6 epitope is not obscured by N-glycan shielding, a viable host recognition of this area is assumed. Indeed, the fact that this area is recognized by F6 and 3BC315, two bNAbs from different clonal lineages and distinct HIV-1 infection background (Fig. 4e), demonstrates that the hosts can effectively appreciate this

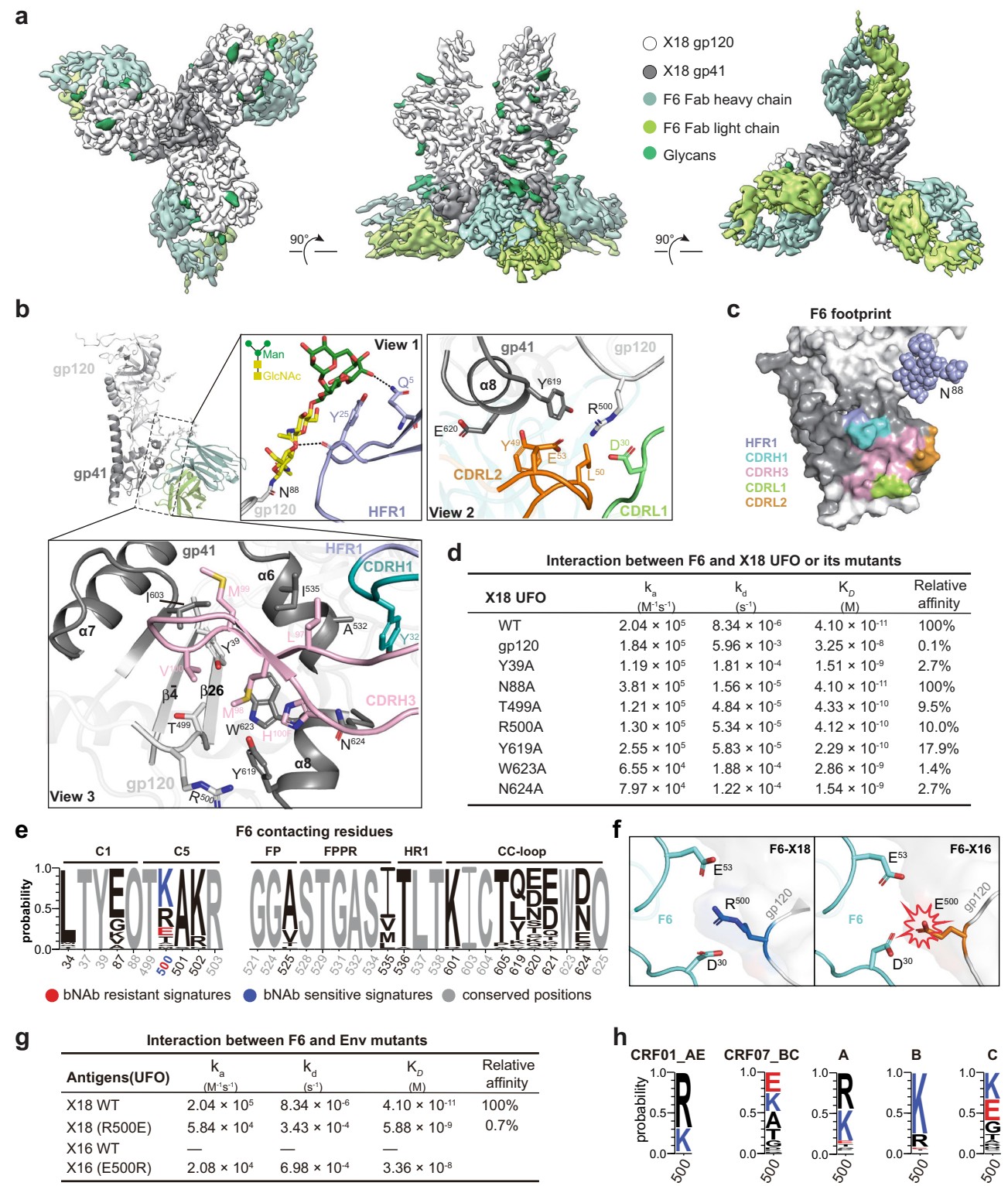

**d**

Interaction between F6 and X18 UFO or its mutants

| X18 UFO | $k_a$ (M⁻¹s⁻¹) | $k_d$ (s⁻¹) | $K_D$ (M) | Relative affinity |
|---|---|---|---|---|
| WT | $2.04 \times 10^5$ | $8.34 \times 10^{-6}$ | $4.10 \times 10^{-11}$ | 100% |
| gp120 | $1.84 \times 10^5$ | $5.96 \times 10^{-3}$ | $3.25 \times 10^{-8}$ | 0.1% |
| Y39A | $1.19 \times 10^5$ | $1.81 \times 10^{-4}$ | $1.51 \times 10^{-9}$ | 2.7% |
| N88A | $3.81 \times 10^5$ | $1.56 \times 10^{-5}$ | $4.10 \times 10^{-11}$ | 100% |
| T499A | $1.21 \times 10^5$ | $4.84 \times 10^{-5}$ | $4.33 \times 10^{-10}$ | 9.5% |
| R500A | $1.30 \times 10^5$ | $5.34 \times 10^{-5}$ | $4.12 \times 10^{-10}$ | 10.0% |
| Y619A | $2.55 \times 10^5$ | $5.83 \times 10^{-5}$ | $2.29 \times 10^{-10}$ | 17.9% |
| W623A | $6.55 \times 10^4$ | $1.88 \times 10^{-4}$ | $2.86 \times 10^{-9}$ | 1.4% |
| N624A | $7.97 \times 10^4$ | $1.22 \times 10^{-4}$ | $1.54 \times 10^{-9}$ | 2.7% |

**g**

Interaction between F6 and Env mutants

| Antigens(UFO) | $k_a$ (M⁻¹s⁻¹) | $k_d$ (s⁻¹) | $K_D$ (M) | Relative affinity |
|---|---|---|---|---|
| X18 WT | $2.04 \times 10^5$ | $8.34 \times 10^{-6}$ | $4.10 \times 10^{-11}$ | 100% |
| X18 (R500E) | $5.84 \times 10^4$ | $3.43 \times 10^{-4}$ | $5.88 \times 10^{-9}$ | 0.7% |
| X16 WT | — | — | — | |
| X16 (E500R) | $2.08 \times 10^4$ | $6.98 \times 10^{-4}$ | $3.36 \times 10^{-8}$ | |

conserved vulnerability despite their diverse genetic background. Meanwhile, the fact that this area is also well captured by vaccination induced immunity in rabbits (1C2) further suggests that this area may be a feasible target site for immunofocusing vaccine design.

**F6 weakens protomers engagement and triggers trimer disassembly**

As mentioned above, the Env apex proximal regions became highly flexible upon F6 binding (Fig. 3a). To find out how F6 triggered this, we compared the structure of X18 UFO in F6-bound state to that in 8ANC195-bound state. In gp41, we observed a 6.3 Å movement of α8 helix and 5.6 Å movement of α6 helix, plus a de-spiral at the C-terminus of α6 in F6-bound state as compared to 8ANC195-bound state (Fig. 5a, close-up views). In 8ANC195-bound state, extensive inter-protomer interactions are observed not only among the central helices (α7), but also between the α9 helix and the α6 helix from the neighbor protomer (Fig. 5b, left). In F6-bound state, although the interactions among α7 helices remain largely unchanged (Supplementary Fig. 6a), the movement of α6 towards F6 and its de-spiral break the hydrophobic (e.g., L595 and aliphatic part of R542*, * denotes elements from neighbor

**Fig. 3 | F6 is a gp120-gp41 interface bNAb. a** Cryo-EM reconstitution of X18 UFO trimer in complex with bNAb F6. Shown are the top (left), side (middle) and bottom (right) views. **b** The interfaces between X18 UFO protomer and F6 Fab are depicted with an overall sideview (left) and detailed in three close-up views (boxed). The gp120 and gp41 subunits of X18 UFO are shown as light and dark grey ribbons, and the HFR1, CDRL1, CDRL2, CDRH1 and CDRH3 of F6 Fab are shown as purple, green, orange, cyan and pink ribbons in close-up views. Glycan N88 and the key residues involved in interactions are shown as stick models. The Env residues are labeled according to HXB2 numbering and CDR residues are labeled following Kabat numbering. **c** Footprint of bNAb F6 on the surface presentation of HIV-1 Env protomer (gp120: white, gp41: grey). Env surfaces that interact with HFR1, CDRH1, CDRH3, CDRL1 and CDRL2 of F6 are colored purple, cyan, pink, green and orange respectively. **d** Interaction kinetics between F6 and X18 UFO trimer (or its indicated

mutants). **e** Amino acid (AA) signatures in bNAb F6 contacts are identified by Fisher's test from the dataset stored in Source Data and displayed in WebLogo plots wherein letter height represents AA frequencies. AAs associated with sensitivity and resistance to F6 are colored blue and red, respectively. "O" denotes an Asn (N) in a PNGS motif. **f** Left: R500 from the gp120 of X18 forms salt bridges with D30 and E53 from the light chain of F6. Right: Change of R500 to E500 would lead to electrostatic repulsion among E500, D30 and E53, thus explaining the failure of X16 to bind F6. **g** Binding of F6 to wild-type X18 UFO is almost abrogated by single R500E mutation, and single E500R mutation renders X16 UFO responsive to F6. **h** WebLogo plots of the AA frequencies at residue 500 in subtypes A ($n = 323$), B ($n = 2322$), C ($n = 1577$), CRF01_AE ($n = 618$) and CRF07_BC ($n = 56$) Envs. Env sequences are from the Los Alamos HIV sequence database (www.hiv.lanl.gov). Source data are provided as a Source Data file.

protomer) and hydrophilic (e.g., E648 and R542*) inter-protomer packings between α7/α9 and α6* helices (Fig. 5b, compare right to left). Meanwhile, the involvement of Env residues Y39*, R500* and I535* in F6 binding disrupts their original interaction with neighbouring α9 helix residues K658, E662 and R655 (Fig. 5b, compare right to left), which, together with the steric clashes rendered by F6 (Supplementary Fig. 6b), destabilize the C-terminus of the α9 helix and lead to further inter-protomer packing loss between α9 and α6* helices (Fig. 5b). To summarize, the F6-induced positional and conformational changes in α6 and α9 helices destabilize the interactions among the Env protomers at the base end (Fig. 5b). Consequently, the F6-bound gp41 trimer displays a 9.2° clockwise rotation and ~5 Å dilation at the base end, as compared to 8ANC195-bound state (Fig. 5a).

In gp120, a large outward movement (relative to the trimer axis) was observed in the β4̄/β26 sheets (Fig. 5c, region boxed in black). Consequently, while the 8ANC195-bound X18 UFO measures ~37 Å between the Cα of T499 on adjacent protomer, the same distance measures ~41 Å in F6-bound state (Fig. 5c, close-up view in trimer context). Notably, the gp120 protomer displays clockwise rotation from the sideview upon F6 binding (Fig. 5d, rotational movement indicated with arrows), which likely transmits the disengagement of protomers near the base (Fig. 5b, c) all the way to the apex and eventually leads to the destabilization of the apex proximal regions (V1/V2/V3 and part of C4, V4, V5). Notably, the V1/V2/V3 regions are where the gp120 protomers engage with each other, and their destabilization would loosen the interaction among gp120 protomers at the apex (Fig. 5e).

Hence, the binding of F6 weakens the inter-protomer packings at both the base and the apex of Env trimer (Fig. 5a–e), leading to the possibility of inducing Env trimer disassembly. To test for such possibility, we then perform comparative negative stain EM (nsEM) analysis of 8ANC195-bound or F6-bound X18 UFO trimers at different time points. In the presence of 8ANC195, ~90% of the particles remain to be intact 8ANC195-bound Env trimers over an 18-h period (Fig. 5f, g and Supplementary Table 3). In contrast, while ~83% F6-bound X18 UFO appears to be intact trimers at the beginning (1-h), the percentage of intact trimers decreases to ~48% by the end of 7 h, and to less than 10% by the end of 18 h (Fig. 5f, g and Supplementary Table 3). Meanwhile, 2D class averages show that most of the particles have disassembled into F6-bound gp120-gp41 protomers by the end of 18 h (Fig. 5f). Notably, F6-induced disassembly was also observed for Q769 Env (subtype A) (Supplementary Fig. 6c). These results clearly demonstrate that F6 binding would first destabilize the apex proximal regions and eventually lead to Env trimer disassembly, and our cryo-EM structure of F6-bound X18 UFO trimers has captured an intermediate state along the path of such induced Env disassembly (Fig. 3a).

### Neutralization mechanism of F6

Next, we sought to reveal the neutralization mechanism of F6. Consistent with the destabilization of trimer apex proximal regions, the Apex bNAbs PGT145 and PG9 recognize the F6-bound Env trimer with

markedly reduced affinities than to the unbound Env (Supplementary Fig. 7a, b). Such observation not only confirms that the binding of F6 is truly destabilizing the trimer apex, but also indicates that destabilization of corresponding regions would hinder the binding of proteins that recognize these regions. Notably, the V1/V2 stem, β20/β21 hairpin and the V5 loop all play critical roles in receptor CD4 engagement[32,45] (Supplementary Fig. 7c–e). Indeed, a single mutation at W427 in β20/β21 hairpin is sufficient to abrogate CD4 binding and render HIV-1 virus non-infectious[46]. Upon F6 binding, the V1/V2 regions (117–208 aa), the β20/β21 hairpin (421–439 aa) and part of the V5 loop (459–463 aa) all became destabilized (Supplementary Fig. 7f). Hence, it is conceivable that F6-induced destabilization of these regions would inevitably damage their interaction with CD4 (Supplementary Fig. 7d). Consistently, drastic affinity decrease was observed between soluble CD4 (sCD4) and Env when F6 is present (Fig. 6a and Supplementary Fig. 7b). Hence, F6-induced destabilization of apex proximal regions would hinder the binding of receptor CD4, thereby representing one potential neutralization mechanisms of F6.

Destabilization of apex proximal region is only the initial event triggered by F6. As time goes by, the disengagement of gp120-gp41 protomers at both the trimer apex and base would eventually lead to the disassembly of Env trimers into gp120-gp41 protomers (Fig. 5), thereby hindering host-viral membrane fusion that relies exclusively on trimeric Env[3]. Notably, 3BC315 and 1C2, the two antibodies that recognize similar areas as F6 on Env surface, also induce trimer disassembly and such irreversible disassociation of Env trimers into gp120-gp41 protomers has been correlated with their greater apparent neutralization potency at longer incubation time[40,41]. We thus also monitored the neutralization potency of F6 over an 18-h period using a similar pre-incubation neutralization assay[40]. For both X18 and Q769 viruses, the apparent IC$_{50}$ of F6 improved 3.8- and 4.2-fold respectively due to F6-induced trimer disassembly over the 18-h incubation, even greater than that of sCD4 (2.6- and 3.4-fold, IC$_{50}$ decreases due to sCD4-induced gp120 shedding) (Fg.6b and Supplementary Fig. 8). Given that 3BC315 has been found to also accelerate gp120 shedding, we checked if F6 could induce the shedding of gp120 from X18 and Q769 viral surfaces as well. No obvious gp120 shedding was observed in the presence of F6 or 35O22, while sCD4 efficiently induces gp120 shedding as a positive control (Fig. 6c and Supplementary Table 4). Hence, F6 binding would lead to the irreversible decay of functional Env trimers into gp120-gp41 protomers on viral surfaces, thereby representing another potential neutralization mechanisms of it.

## Discussion

As a prototypical 'evasion-strong' pathogen, the HIV-1 virus represents an insurmountable obstacle for classical vaccine approach[9]. As such, a more systematic vaccine strategy that involves templating the 'conserved immune vulnerabilities' learnt from HIV-1 bNAbs onto a consensus Env framework has elicited an upsurge of interest in structure-based HIV-1 vaccine design, which nevertheless entails a

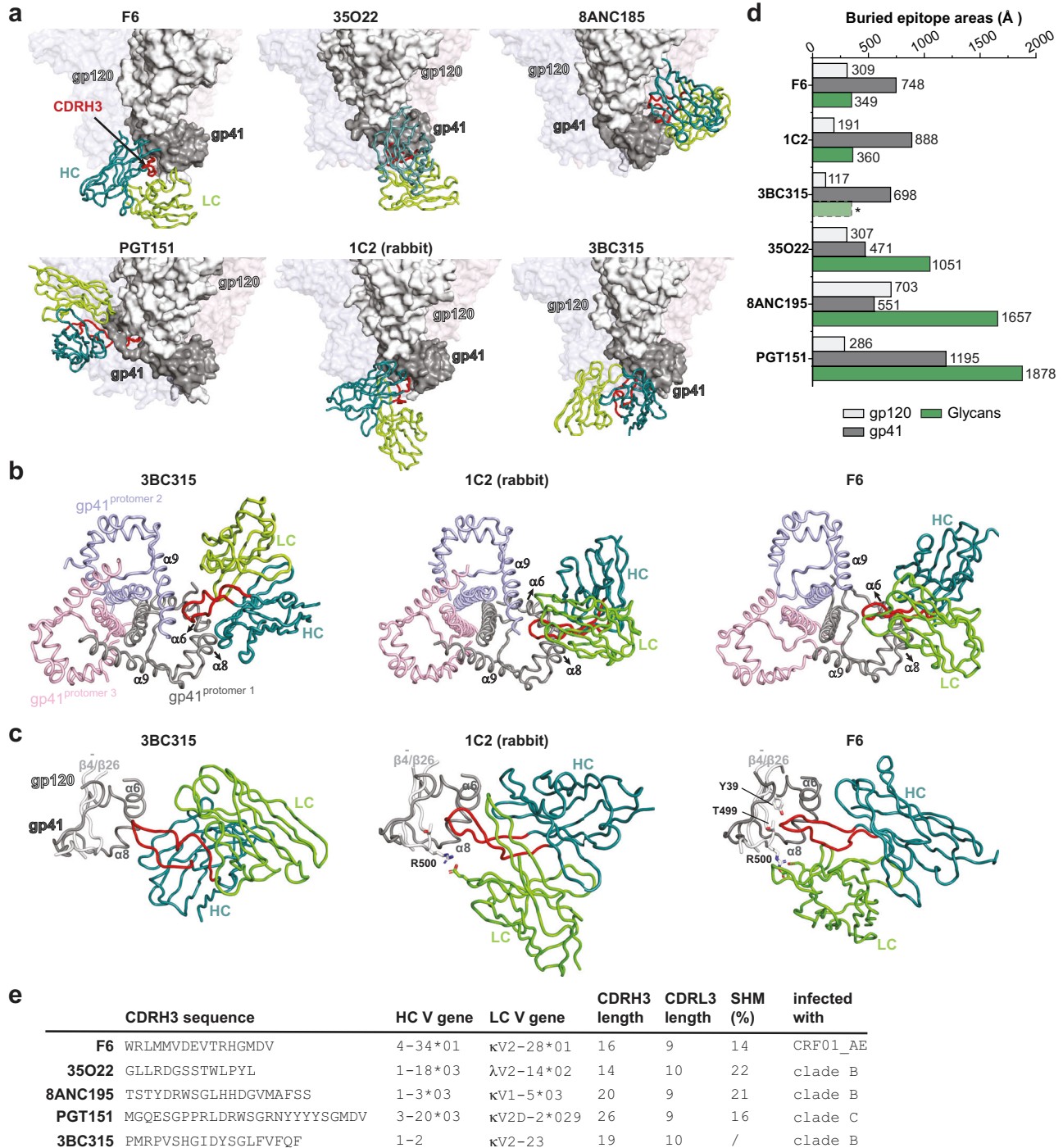

**Fig. 4 | Epitope comparison of gp120-gp41 interface bNAbs. a** The binding mode of F6 is compared to those of other interface bNAbs. The structures of F6-Env and 8ANC195-Env complexes are from this study, the structures of 35O22-Env, PGT151-Env and 1C2-Env complexes are based on PDB: 4TVP, 5FUU and 6P65[25, 38, 41]. The structures of 3BC315-Env are obtained by fitting the 3BC315 crystal structure (PDB: 5CCK) and the BG505 Env model (PDB: 4TVP) into the 9 Å cryo-EM map of 3BC315 in complex with BG505 SOSIP (EMD-3067)[40]. In each panel, only the variable regions of one bNAb are shown, the gp120 and gp41 of the major interacting Env protomer are shown as white and grey surface presentations while the other two protomers are rendered transparent for clarity. **b** Comparison of the interactions between gp41 and 3BC315, 1C2 and F6. 3BC315 and 1C2 interact with the α6 and α8 from one protomer (grey), as well as α9 from the neighboring protomer (purple). F6 only

interact with the α6 and α8 from the same protomer. **c** Comparison of the interactions between gp120 and 3BC315, 1C2 and F6. 3BC315 barely interacts with gp120 residues. The light chain (LC) of 1C2 makes limited interaction with R500 from gp120. The light chain (LC, CDRL1 and CDRL2) and heavy chain (HC, CDRH3) of F6 make extensive and crucial interactions with R500, T499 and Y39 from gp120. Note that the CDRH3 of F6 inserts much deeper to reach β4̄/β26 (gp120) than that of 3BC315 or 1C2. **d** Bar graphs denoting gp120, gp41 and glycan interacting surface areas on Env for each indicated bNAb. *The contribution of glycans to 3BC315 epitope is inferred rather than calculated from the docking model. **e** The CDRH3 sequences, heavy and light chain V gene usage, CDRH3 and CDRL3 lengths, somatic hypermutation (SHM) rates and the origins of the indicated gp120-gp41 interface bNAbs. Source data are provided as a Source Data file.

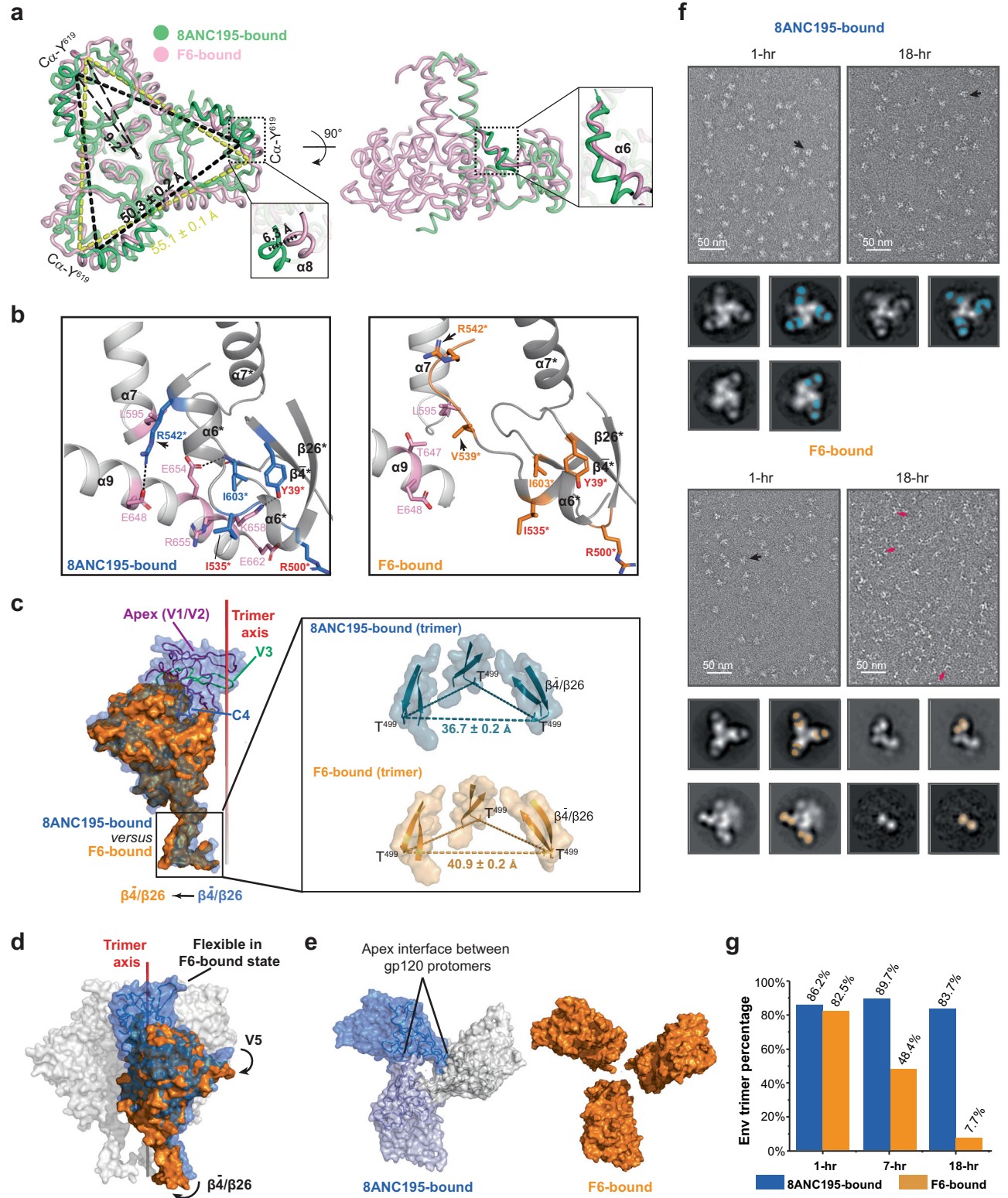

comprehensive structural and antigenic understanding of Envs from diverse genetic background, i.e., subtypes.

CRFs are products of inter-subtype genomic recombination events in dually infected individuals and represent the dominant subtypes in Southeast and East Asia[16]. Despite the growing prevalence of CRFs globally[47], a structural understanding of the antigenic landscape of their Envs remains lacking. Here, we present cryo-EM structures of 8ANC195-bound CRF01_AE and CRF07_BC Envs at 3.0 Å and 3.5 Å resolutions, respectively. The Envs of these two CRFs exist in

closed, prefusion conformation and manifest an overall structural homology to Envs from other subtypes[23–27], thereby supporting the concept of consensus Env scaffold for immuno-vulnerability grafting (Fig. 1a–c). Structural and mass spectrometry visualization of the glycan shield on X18 UFO and X16 UFO, the representative CRF01_AE and CRF07_BC Envs, highlights a highly glycosylated V1 region of X18 UFO (Fig. 1e). Such observations, together with cross-clade statistical analyses of 4896 Env sequences, lead to the discovery that the V1 loops of subtype CRF01_AE are significantly longer and glycosylated more

**Fig. 5 | F6 binding weakens protomers engagement and triggers trimer disassembly. a** Left: bottom view of aligned gp41 trimers, gp41 trimer displays a 9.2° clockwise rotation and a -5 Å dilation in F6-bound state (pink) as compared to 8ANC195-bound state (green). Right: side view. The two boxed close-up views depict the -6 Å translocation of α8 helix and the de-spiral of α6 helix in F6-bound state as compared to 8ANC195-bound state. **b** F6 Binding destabilizes the interactions among Env protomers at the base end of gp41. Residues involved in inter-protomer packings in 8ANC195-bound Env trimer are shown as pink (protomer 1) and blue (protomer 2) sticks (left). The same residues are shown as pink and orange sticks in F6-bound state (right). F6-binding Env residues are labeled red. Arrows indicate residues displaying large positional changes in F6-bound state. Structural elements and residues of protomer 2 are denoted with * and electrostatic interactions are indicated with black dashes. **c** Aligned gp120 protomers highlight the outward movement of β4̄/β26 (indicated with an arrow) and the missing of apex proximal regions (V1/V2/V3 and part of C4, V4, V5) in F6-bound state (orange) as compared to 8ANC195-bound state (blue). Box: close-up views of the β4̄/β26 sheets in trimer context. **d** Sideview of aligned gp120 trimers in surface representations suggests the possible transmission route of conformational changes. F6-binding induced rotational movements in β4̄/β26 and V5 regions are indicated with black arrows. **e** Left: the V1/V2 region from different gp120 protomers interacts with each other in 8ANC195-bound state. Right: the same interface is missing in F6-bound state due to the F6-binding induced destabilization of apex proximal regions. **f** nsEM micrographs of 8ANC195-bound and F6-bound X18 UFO at 1-h and 18-h. Representative Fab-bound Env trimers and Fab-bound gp120-gp41 protomers are pointed with black and red arrows respectively. Shown below each nsEM micrograph are the representative 2D class averages of corresponding samples and the locations of Fabs in each class are highlighted with blue (8ANC195) or orange (F6) blocks on the right. **g** Percentage of intact Env trimers under nsEM at the indicated time points. Data for this plot are shown in Supplementary Table 3.

heavily than V1 loops from other subtypes (Fig. 1g–i). We further correlated the length and glycosylation level of V1 with bNAbs sensitivities and revealed the inverse correlations between these two molecular features and the viral sensitivities to representative V3 and Apex bNAbs (Fig. 2). Thus, CRF01_AE viruses are more likely to be resistant to V3 or Apex bNAbs. Moreover, a PNGS shift from N332 to N334 in more than 96% of CRF01_AE viruses would further exacerbate such resistance to V3 bNAbs[12], given the key role of N332 in V3 bNAbs epitopes. The above observation has important implications for both immunofocusing vaccine development and the therapeutic or prophylactic usage of corresponding bNAbs. For instance, clinical trials of HIV-1 vaccines designed to focus immune response to conserved V3 or Apex vulnerabilities should avoid CRF01_AE prevailing geographical regions, as such vaccines are more likely to fail in preventing CRF01_AE infections. As for the prophylactic application of bNAbs, when receiving combinations of bNAbs before traveling, regional distribution of HIV-1 subtypes in destination should be carefully considered for selecting the optimal bNAbs[48].

To further extend our antigenic understanding of CRF Envs, we solved a 4.1 Å cryo-EM structure of CRF01_AE Env in complex with F6, the first reported bNAb isolated from CRF01_AE-infected individual[20]. We found that F6 recognizes a gp120-gp41 spanning structural epitope (Fig. 3), which is distinct from that defined by bNAbs 8ANC195, 35O22 and PGT151 but overlaps that of 3BC315 and rabbit antibody 1C2 (Fig. 4). Previously, V1/V2 apex, V3 or CD4bs bNAbs have all been found to frequently target highly overlapping epitopes, although they were identified from different donors and belong to different clonal lineages (Supplementary Fig. 9). In contrast, epitopes of gp120-gp41 interface bNAbs do not seem to converge except for bNAbs that bind FP. Here, the identification of F6 epitope and its similarity to 3BC315 and 1C2 epitopes highlight that the glycan free area near their epitopes, akin to the CD4bs, V3 glycan supersite, V1/V2 apex and FP, is also a repeatedly targeted immune vulnerability on the Env surface. Moreover, the observation that this area is well captured by vaccination induced rabbit antibody 1C2 further supports the application potential of this area in immunofocusing vaccine design.

Interestingly, F6 neutralizes through a unique dual mechanism. Upon F6 binding, the apex proximal regions of Env get destabilized, which in turn hinders the binding of CD4 receptor, representing one potential neutralization mechanisms of F6 (Fig. 6d, F6-bound intermediate state). The decreased binding of CD4 to F6-bound Env is reminiscent of 8ANC195, whose binding to Env also hinders the engagement of CD4[28]. While 8ANC195 constrains Env in a conformation that is incompatible with CD4 binding, F6 rather works by allosterically changing the CD4 binding surface on Env. Besides destabilizing the trimer apex proximal region, F6 binding also triggers protomer disengagement at the trimer base (Fig. 5a, b). Together, these conformational changes eventually lead to trimer disassembly (Fig. 6d, F6-bound final state). As disassembled Env trimer loses its

ability to form fusogenic six-helical bundle (6HB) properly, efficient host-viral membrane fusion cannot be mediated. Hence, Env-disassembling ability likely represents the other neutralization mechanism of F6. Notably, induced disassembly of Env trimers has also been observed for 3BC315 and 1C2[40,41], as well as non- or weakly-neutralizing antibodies elicited by subunit immunogens in vaccine studies[49]. The latter antibodies approach the Env at an angle that is incompatible with the membrane and thus would not be able to disassemble membrane-embed Envs on virions, explaining their lack of neutralizing activity[49]. Besides inducing trimer decay, 3BC315 has also been shown to neutralize through accelerating gp120 shedding[40]. We thus also checked if F6 could induce the shedding of gp120 from pseudotyped HIV virions and observed no obvious shedding (Fig. 6c and Supplementary Table 4). Such difference in the neutralization mechanisms of F6 and 3BC315 is consistent with their similar and different epitopes, as F6 makes extensive and key interactions with gp120 while 3BC315 hardly interacts with gp120 except for the N88 glycan[40] (Fig. 4c).

In summary, here we depicted the structural and antigenic landscape of the Envs from two Asia-prevalent subtypes, CRF01_AE and CRF07_BC. We discovered unique molecular features of CRF01_AE V1 region and further revealed that these features are associated with decreased sensitivities to certain bNAbs. Furthermore, by solving the structure of CRF01_AE Env in complex with bNAb F6, we unraveled a bona fide site of vulnerability on Env and elaborated the dual neutralization mechanism of F6. We envision that the information obtained from this study will provide valuable clues for immunofocusing vaccine development and the clinical usage of related bNAbs.

## Methods

### Cell lines and cell cultures

All mammalian cell lines were incubated at 37 °C in a humidified atmosphere containing 5–8% $CO_2$. FreeStyle™ 293-F suspension cells (Gibco, R79007) were maintained in FreeStyle 293F expression medium (Gibco, 12338-018), ExpiCHO-S™ cells (Gibco, A29127) were maintained in ExpiCHO™ Expression Medium (Gibco, A29100-01), TZM-bl cells (NIH AIDS Research and Reference Reagent Program, Cat. no. 8129) were maintained in DMEM medium (Gibco,11995-065). Sf9 (Gibco, 11496015) and High Five Cells (Gibco, B85502) were maintained in Sf-900™ II SFM medium (Gibco,10902-088) and ESF 921 medium (Expression Systems, 96-001-01) at 27 °C respectively.

### Gene cloning, protein expression and purification

X18 UFO and X16 UFO were constructed from the ectodomain (30-660 aa) of X18 (GenBank: AIS42911.1) and the ectodomain (29-661 aa) of X16 (GenBank: AIS42666.1) following UFO design[22]. Specifically, A501C and T605C mutations were introduced, furin cleavage site was replaced with 2×GGGGS linker, and sequences 'IVQQQSNLLRAIEAQQ HLLQL' in X18 HR1 region and 'IVQQQNNLLRAIEAQQHMLQL' in X16 HR1 region

**a**

| Binders | Envs | $k_a$ (M$^{-1}$s$^{-1}$) | $k_d$ (s$^{-1}$) | $K_D$ (M) | Relative affinity |
|---|---|---|---|---|---|
| **sCD4** | X18 UFO | $4.22 \times 10^3$ | $4.10 \times 10^{-5}$ | $9.72 \times 10^{-9}$ | 100% |
| | X18 UFO + F6 (RT, 1-hr) | $1.56 \times 10^3$ | $9.30 \times 10^{-5}$ | $5.96 \times 10^{-8}$ | 16.3% |
| **sCD4** | Q769 UFO | $5.82 \times 10^4$ | $7.01 \times 10^{-5}$ | $1.21 \times 10^{-9}$ | 100% |
| | Q769 UFO + F6 (RT, 1-hr) | $1.92 \times 10^4$ | $2.64 \times 10^{-4}$ | $1.38 \times 10^{-8}$ | 8.8% |

**b**

| X18 (CRF01_AE, Tier 2) | | | | | |
|---|---|---|---|---|---|
| Preincubation (hr) | IC50 (µg ml$^{-1}$) | Fold change | IC50 (µg ml$^{-1}$) | Fold change | IC50 (µg ml$^{-1}$) | Fold change |
| | 35O22 | | sCD4 | | F6 | |
| 1 | 0.021 | | 3.87 | | 40.87 | |
| 6 | 0.014 | 1.5 | 2.30 | 1.7 | 36.86 | 1.1 |
| 18 | 0.011 | 1.9 | 1.49 | 2.6 | 10.76 | **3.8** |

| Q769 (Clade A, Tier 2) | | | | | |
|---|---|---|---|---|---|
| Preincubation (hr) | IC50 (µg ml$^{-1}$) | Fold change | IC50 (µg ml$^{-1}$) | Fold change | IC50 (µg ml$^{-1}$) | Fold change |
| | 35O22 | | sCD4 | | F6 | |
| 1 | 0.017 | | 2.25 | | 6.06 | |
| 6 | 0.016 | 1.1 | 1.04 | 2.2 | 2.82 | 2.2 |
| 18 | 0.011 | 1.5 | 0.66 | **3.4** | 1.45 | **4.2** |

**c**

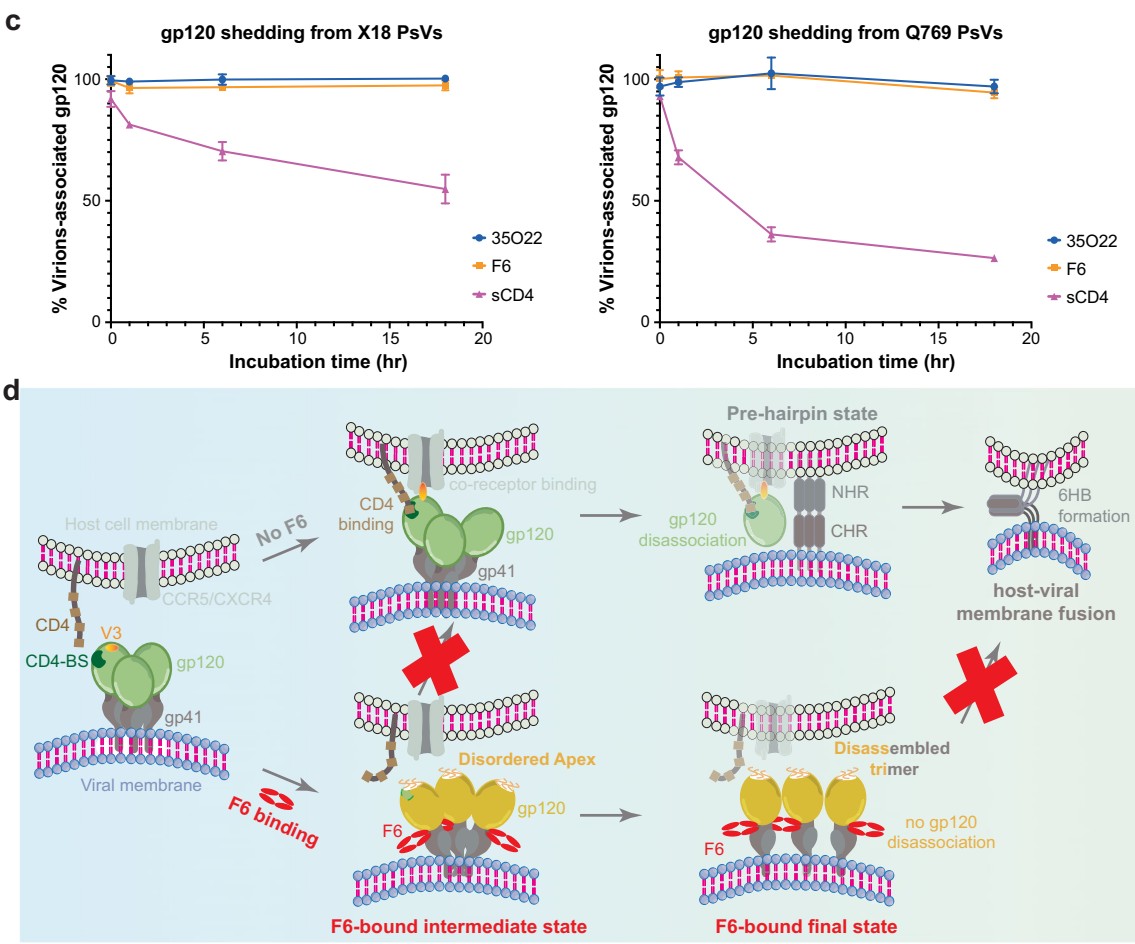

were replaced with optimized sequence 'NPDWLPDM'. Q769 UFO was constructed from the ectodomain (29-660) of Q769_d22 (GenBank: AAM66234.1) as described above. Coding sequence of X18 UFO, X16 UFO or Q769 UFO were then cloned into a modified pCMV-mam vector (or pcDNA3.4 vector) with a N-terminal t-PA signal peptide. Mutants of X18 UFO and X16 UFO were constructed by site-directed mutagenesis and verified by DNA sequencing. Oligos used for plasmids

construction are summarized in Supplementary Table 5. Expression and purification of the wild-type proteins and mutants were performed using the method described below.

X18 UFO and X16 UFO Env trimers used for cryo-EM and mass spectrometry studies were produced by transient transfection of ExpiCHO-S cells according to the manufacturer's protocol. Six days post transfection, Env trimers were captured from clarified cell culture

**Fig. 6 | Neutralization mechanisms of F6. a** Interaction kinetics between sCD4 and Envs or F6-bound Envs. For the measurement of the latter, X18 UFO or Q769 UFO was only incubated with F6 for 1 h to preclude the interference of F6-induced trimer disassembly (F6-induced trimer disassembly is not prominent at 1-h time point). **b** Pseudotyped X18 or 769 virus was incubated with 35O22 IgG, sCD4 or F6 IgG for indicated periods, before being added to TZM-bl cells. Fold change (decrease) in $IC_{50}$ is calculated by normalizing against the $IC_{50}$ of standard 1-h incubation. $IC_{50}$ values of F6 decrease significantly during the prolonged incubation, indicating that Envs are disabled by F6 over time, which is consistent with F6-induced trimer disassembly. **c** Pseudotyped X18 or 769 virions were incubated alone or with 35O22, sCD4 or F6 for indicated periods. Incubated virions were pelleted and gp120 remaining associated with virions was detected by ELISA. The amount of gp120

remaining associated with antibody treated virions are normalized against the untreated sample (100%). sCD4 and 35O22 serve as positive and negative control respectively. Data are shown as means ± SD ($n = 3$ biologically independent experiments). Underlying data for this panel are given in Supplementary Table 4. **d** Schematic diagram illustrating the dual neutralization mechanism of F6: (1) F6 binding first destabilizes the apex proximal regions of Env (F6-bound intermediate state), thereby hindering the binding of CD4 receptor; (2) F6-induced destabilization of trimer apex and gp120-gp41 protomers disengagement at the trimer base finally lead to the disassembly of Env trimer into gp120-gp41 protomers (F6-bound final state), disassembled Env trimer loses its ability to form fusogenic six-helical bundle (6HB) and thus cannot mediate efficient host-viral membrane fusion.

supernatants using *Galanthus nivalis* lectin (GNL) resin (Vector Labs) and elution fractions were then further purified on a Superpose 6 10/300 GL column (Cytiva) in buffer containing 25 mM HEPES pH 7.5, 250 mM NaCl. Gel filtration peak fractions containing the Env trimers were then concentrated and stored at 4 °C until further use.

Wild type X18 UFO, X16 UFO and Q769 UFO or their mutants used for SPR experiments were produced by transient transfection of FreeStyle 293F suspension cells with polyethylenimine (PEI) (Yeasen). Cells were transfected at 0.8–1.2 million cells/ml and harvested 5–6 days later, and proteins were purified and stored in similar ways as described above.

X18 SOSIP-His, X16 SOSIP-His were designed and constructed according to BG505 SOSIP design[30]. BG505 SOSIP-His, JRFL SOSIP-His, 16055 SOSIP-His constructs were synthesized by Tsingke Biotechnology (Shanghai, China) according to published sequences. All these Env trimers in SOSIP design were co-expressed with furin protease in FreeStyle 293F suspension cells. The Env proteins were harvested from the supernatant one week later and captured by a PGT151 affinity column, which was made in house by immobilizing PGT151 IgG on a HiTrap NHS-activated HP column (Cytiva) according to manufacturer's protocol. Further purification was performed with a Superdex 200 10/300 column (Cytiva). Purified SOSIP trimers were concentrated and stored in buffer containing 25 mM HEPES pH 7.5, 300 mM NaCl at 4 °C until further use.

Genes of Heavy chains and Light chains of VRC01, PG9, ACS202 and PGT145 Fabs were synthesized by Tsingke Biotechnology (Shanghai, China) or GenScript Biotech Corporation (Nanjing, China) and cloned into pcDNA 3.4 expression vector (Invitrogen). Genes of F6 Fab variable regions were obtained in similar ways and cloned into modified pFUSE vectors (InvivoGen). The 8ANC195 Fab, 35O22 Fab, PGT151 Fab and PGT121 Fab recombinant mammalian expression plasmids were kind gifts from Prof. Ian A. Wilson at the Scripps Research Institute. Genes of F6 and 35O22 in IgG1 format were synthesized by GenScript Biotech Corporation (Nanjing, China) and cloned into pcDNA3.1 expression vector.

All Fabs used in SPR or gp120 shedding experiments and IgGs used in neutralization assays were expressed through co-transfecting heavy and light chain plasmids (1:1 molar ratio) with polyethylenimine (PEI) (Yeasen) into FreeStyle 293F suspension cells. The transfected cells were cultured for another 6 days before being harvested. Antibodies in the medium were captured with a lambda affinity column (Cytiva) or HiTrap Protein A or G HP column (Cytiva) and further purified by size exclusion chromatography. Purified antibodies were concentrated and stored in PBS buffer (pH 7.4) at −80 °C until further use.

F6 Fab used for X-ray crystallography or cryo-EM was produced by transient transfection of ExpiCHO-S cells according to the manufacturer's protocol and purified as described above. Purified F6 Fab was concentrated and stored in buffer containing 10 mM Tris-HCl pH 7.5, 100 mM NaCl at −80 °C until further use. 8ANC195 Fab used for cryo-EM sample preparation and PGT151 IgG for PGT151 affinity column preparation were expressed in insect cells using the Bac-to-Bac system

(Gibco). Briefly, heavy chains and light chains of 8ANC195 Fab or PGT151 IgG were inserted into a modified pFastBac dual vector (Gibco) to generate pFastBac dual-8ANC195-Fab-3C-His and pFastBac dual-PGT151-IgG-3C-His plasmids. Gene of sCD4 (D1 and D2 domains) was synthesized by Tsingke Biotechnology (Shanghai, China) and inserted into a modified pFastBac1 vector (Gibco) with a C-terminal 3C protease cleavage site followed by 10× His tag. Baculoviruses were prepared by transfecting freshly prepared bacmids into Sf9 cells using FuGENE HD (Promega) and used to infect suspension High Five cells at a MOI of 5 to 10. Medium supernatant of infected High Five cells was harvested after 72 h and supplemented with 1 mM PMSF before being loaded onto an Excel Ni-NTA column (Cytiva). Elution fractions containing 8ANC195 Fab or PGT151 IgG were then pooled and further purified with a HiTrap Protein G HP column (Cytiva) or HiTrap Protein A HP column (Cytiva) followed by gel filtration chromatography on a Superdex 200 10/300 column (Cytiva) equilibrated with 25 mM HEPES pH 7.5, 150 mM NaCl. Elution fractions containing sCD4 were cleaved by 3C protease overnight at 4 °C before being loaded onto Ni-NTA column again. sCD4 proteins in flow through were then collected and further purified by gel filtration chromatography on a Superdex 75 10/300 column (Cytiva) equilibrated with 25 mM Tris-HCl pH 7.5, 150 mM NaCl. Purified antibodies and sCD4 were concentrated and stored at −80 °C until further use.

### Surface plasmon resonance (SPR)

Single-cycle kinetics experiments were performed with a Biacore 8 K apparatus (Cytiva). All assays were performed with a running buffer containing 20 mM HEPES pH 7.5, 300 mM NaCl, 3 mM EDTA and 0.01% v/v Tween-20 at 25 °C. Injection parameters of each analyte were adjusted according to pilot binding assay results. The resulting sensorgrams were fitted using 1:1 Langmuir binding model within the Biacore Insight Evaluation Software (Cytiva).

To determine the binding affinity of F6 to X18 UFO and X16 UFO (or their mutants), Env proteins were each immobilized to a single flow cell on a CM5 sensor chip (Cytiva) using an amine coupling kit. After three injections of running buffer, twofold dilutions of F6 Fab were injected over both the sample and reference flow cells at flow rate 30–50 μL/min with a contact time of 120 s and a dissociation time of 400–3200 s. To measure the binding affinity of six representative Fabs (35O22, 8ANC195, ACS202, VRC01, PGT121, PG9) to five Envs from different subtypes (X18 SOSIP-His, X16 SOSIP-His, BG505 SOSIP-His, JRFL SOSIP-His, 16055 SOSIP-His), each Env protein was captured onto Series S NTA sensor chips (Cytiva) via $Ni^{2+}$/NTA chelation. Running buffer was injected at flow rate 50 μl/min for 5 min followed by twofold dilutions of each Fab. To compare the binding affinities of sCD4, PG9 or PGT145 to X18 UFO and F6-bound X18 UFO, anti-histidine antibody was primary amine-coupled onto a CM5 Sensor Chip (Cytiva) at a coupling density of 14,000 resonance units (RUs). Capture proteins (His-tagged X18 UFO alone or X18 UFO-F6 complex) were then immobilized as ligand. After three injections of running buffer, twofold dilutions of sCD4, PG9 or PGT145 were injected as analytes. To compare the binding affinities of sCD4 to Q769 UFO and F6-bound Q769

UFO, His-tagged sCD4 was captured onto anti-histidine antibody coupled-CM5 Sensor Chip (Cytiva), and twofold dilutions of Q769 UFO or F6-saturated Q769 UFO were injected as analytes.

### Crystallization and X-ray structure determination

Crystallization screening of F6 Fab was carried out using the hanging drop vapor diffusion method at 16 °C. Diffraction-quality crystals of F6 were obtained in conditions containing 0.2 M sodium citrate, 20% PEG 3,350. Crystals were harvested with 20% (v/v) glycerol as cryoprotectant and flash-cooled in liquid nitrogen for data collection. X-ray diffraction data were collected with Blu-Ice 5 program on beamline BL19U1 at the Shanghai Synchrotron Radiation Facility (SSRF) and processed using HKL2000 program[50]. Structure was solved through molecular replacement (MR) with PHASER[51], using MR templates derived from PDB deposit 5BMF[52]. Iterative model building and refinement were carried out in COOT and PHENIX respectively[53,54]. The quality of the final model was analyzed with MolProbity[55]. Data collection and refinement statistics are outlined in Supplementary Table 2.

### Negative stain sample preparation and data acquisition

X18 UFO-8ANC195, X16 UFO-8ANC195 and X18 UFO-F6 protein complexes to be examined were diluted to an appropriate concentration (0.01-0.03 mg/mL) and dropped onto a freshly glow-discharged carbon-coated grid. After rinsing twice with buffer (25 mM HEPES, 300 mM NaCl, pH 7.5), the grid was stained with 2% uranyl formate (pH 7.0) and then loaded onto a 120 kV TEM for examination.

To visualize the F6-induced disassembly of X18 UFO and Q769 UFO, X18 UFO-8ANC195, Q769 UFO-8ANC195, X18 UFO-F6 and Q769 UFO-F6 protein complexes sampled after 1, 7 and 18 h incubation at room temperature were diluted to an appropriate concentration (0.01–0.03 mg/mL) and dropped onto a freshly glow-discharged carbon-coated grid. After rinsing twice, the grid was stained with 2% uranyl formate (pH 7.0) and then loaded onto a 120 kV TEM for examination. 15–71 representative images for each sample at different time points were manually collected at 73,000 × nominal magnification (1.96 Å/pixel) with a defocus range of −2 to −3 μm, using a Talos C-Twin electron microscope (Thermo Fisher Scientific) operated at 120 kV and equipped with a 4 K × 4 K Ceta CCD camera.

### Negative stain image processing and quantification of trimer disassembly

Contrast transfer function parameters were estimated using CTFfind4.1 and particles were automatically picked with crYOLO using a general model[56,57]. Subsequent image processing was performed in RELION v3.1[58]. Particles were initially extracted with a box size of 300 pixels and sorted by reference-free 2D classification[58]. Particles resembling an Env trimer or Env protomer (with or without bound Fabs) were selected for subsequent calculation. The percentage of intact Env trimers was obtained by dividing the number of all particles with the number of intact Env trimer particles at that time points.

### Cryo-EM sample preparation

Complexes of X18 UFO-8ANC195, X16 UFO-8ANC195 and X18 UFO-F6 were assembled by incubating corresponding Fabs with Env trimers at a 4:1 molar ratio overnight (or <3 h for X18 UFO-F6) at 4 °C. Excessive antibodies were removed by gel filtration chromatography on Superose 6 10/300 (GE Healthcare). Peak fractions containing the Env-Fab complexes were immediately pooled, concentrated, and subjected to cryo-EM sample preparation. Prior to grid preparation, Env-Fab complexes at 0.8–1 mg/ml (or 8 mg/ml for X18 UFO-F6) in 25 mM HEPES pH 7.5, 250 mM NaCl were supplemented with small amounts of n-Dodecyl β-D-Maltoside (DDM, Anatrace) and Cholesteryl Hemisuccinate Tris Salt (CHS, Sigma) to solve the orientation preference issue. 3 μl protein sample was added to freshly glow-discharged

(Solarus Gatan Plasma System $H_2/O_2$, 30 s) holey carbon grids (Quantifoil Cu R 0.6/1.0). After blotting for 2.5–4.5 s with TED PELLA 595 filter paper at 281 K and 100% relative humidity, sample grids were immediately vitrified in 100% liquid ethane using a Mark IV Vitrobot (Thermo Fisher Scientific).

### Cryo-EM data collection, image processing and refinement

Single particle cryo-EM data were collected with SerialEM software on a Titan Krios electron microscope (Thermo Fisher Scientific) operating at 300 kV, equipped with a Gatan K3 Summit direct electron detector for data acquisition. Movies were collected at a nominal magnification of 22,500× (0.53 Å/pixel) using a defocus range of 0.8–2.5 μm. The exposure time was set to 2.8 s with dose rate of 20 e⁻/pixel/s, resulting in a total dose of 50 electrons per Å².

Movies for X18 UFO-8ANC195 were motion-corrected and dose-weighted by MotionCorr2 frame alignment program in RELION3.1[58]. The contrast transfer function (CTF) for each aligned micrograph was estimated by CTFfind4.1[56]. 3050 images were inspected manually and micrographs with poor CTF fits or drifts were discarded. A total of 1,156,571 particles were automatically picked from 2952 dose-weighted micrographs using 2D class references from 3553 manually picked particles. After two rounds of 2D classification, 806,307 particles were selected and extracted for 3D classification with a 3D reference map quickly built by cryoSPARC v3[59]. Particles from the top two classes were combined and subjected to another round of 3D classification using a soft mask in which the constant regions of 8ANC195 Fab were masked out. Particles of the best-classified class were subjected to CTF refinement and particle polishing before final 3D refinement and post-processing, resulting in a 2.99 Å map. Movies for X16 UFO-8ANC195 were processed in similar ways, yielding a 3.49 Å map.

Movies for X18 UFO-F6 were aligned using multi-patch motion correction, followed by CTF estimation by multi-patch CTF estimation in cryoSPARC v3[59]. A total of 3,691,481 particles from 6911 images were picked and subjected to several rounds of 2D classification in cryoSPARC v3[59]. After one round of 3D ab initio reconstruction and heterogeneous refinement, two major classes containing 350,799 particles were selected and combined for homogeneous refinement with C1 symmetry, yielding a 4.48 Å map. Non-uniform refinement using C3 symmetry then improved the overall resolution to 4.14 Å (Supplementary Fig. 4c). To further improve the resolution at the interface, local refinement with a soft mask (encompassing the variable regions of F6 Fab and V1/V2/V3-omitted X18 UFO protomer) and symmetry expansion were performed, yielding a 4.05 Å map (Supplementary Fig. 4d).

### Cryo-EM model building and refinement

The structure of BG505 SOSIP in complex with 8ANC195 Fab (PDB: 5CJX) was fitted into the X18 UFO-8ANC195 EM density map denoised by DeepEMhancer[60] with the ChimeraX[61] program and then rebuilt in Coot[54] to generate the initial model for X18 UFO-8ANC195. The initial model of X16 UFO-8ANC195 was obtained in similar ways. An initial model for X18 UFO-F6 was obtained by manually fitting the gp120 and gp41 domains from the X18 UFO-8ANC195 structure and the crystal structure of F6 Fab into the EM density with ChimeraX[61]. All resulting models were subjected to real-space refinement in PHENIX[53]. Glycans were manually built using the Carbohydrate module in Coot and refined into the EM density map using Rosetta refinement[62]. The programs Molprobity[55] and EMRinger[63] were used to validate the final models. Cryo-EM data collection, model refinement and validation statistics are outlined in Supplementary Table 1.

### Structural analysis and figure generation

Cα root-mean-square deviation (RMSD) values between different Env Trimers or protomers were calculated using the alignment algorithm of PyMOL with the default 2σ rejection criterion and five

iterative alignment cycles. Buried surface area calculations were performed using PDBePISA (www.ebi.ac.uk/pdbe/pisa) and stored in Source Data Fig. 4d. Structural visualization and figure generation were performed using ChimeraX[61] and PyMOL (Schrodinger). The schematic diagram illustrating the neutralization mechanism of F6 was drawn using Figdraw tools (www.figdraw.com) and Adobe Illustrator (Adobe).

### N-linked glycosylation sites (NGS) identification by mass spectrometry

To identify NGS on X18 UFO and X16 UFO, Env proteins purified from ExpiCHO-S cells were deglycosylated with PNGase F (New England Biolabs) according to the manufacturer's protocol. Two micrograms deglycosylated X18 UFO and X16 UFO proteins were sampled separately with SDS-PAGE and subjected to in-gel trypsin digestion. The desalted digests were then loaded onto a C18 column (Ion optics, 25 cm × 75 μm, 1.6 μm) and subsequently eluted over a 90 min gradient from the column to a Orbitrap Fusion Mass Spectrometer (Thermo Fisher) equipped with Easy LC 1200 and nanoflex spray source. A short wash and blank run were performed between each sample to minimize any carryover. Data were collected in the positive-ion mode in a data-dependent fashion. Briefly, a profile scan ($m/z$ 335-1800 at a resolving power of 60 K) was first conducted followed by the MS/MS of ten most intense precursor ions. Only those ions with a charge state from 2 to 7 were selected for MS/MS scan. High collision activation (HCD) was employed to generate fragment ions. The resolution of MS/MS scan was set to 15 K. The AGC and the maximum injection time of MS2 were set to 1e5 and 50 ms, respectively. All MS/MS spectra were analyzed using the Proteome Discoverer Software (ThermoFisher). Peptides with N-linked glycosylation was identified by searching for a 0.9840 Da mass shift (N-to-D conversion rendered by PNGase F deglycosylation) set as variable modification. Potential N-glycan peptides were manually validated from $MS^2$ data to ensure major fragmentation ions (b and y ions) were observed (see Source Data Fig. 1e).

### Cross-clades analysis of V1 features

Sequences of X16 UFO, X18 UFO, BG505 SOSIP, JRFL SOSIP and 16055 NFL were aligned with HXB2 as reference sequence by HIV Align program using a hidden Markov model in HIV database (https://www.hiv.lanl.gov). The sequence alignment result was displayed with ESPript 3.0 (https://espript.ibcp.fr).

For statistical analysis of subtypes A, B, C, CRF01_AE and CRF07_BC V1 loop features, a filtered web alignment containing 6599 M Group (clades A-K plus recombinant forms) HIV-1 Env sequences were download from HIV database (https://www.hiv.lanl.gov). The V1 region (131-156 according to HXB2 numbering) of each Env sequence was then extracted with Jalview program[64]. 6565 sequences (including the HXB2 reference sequence) were kept after removing redundant Env sequences and Env sequences containing irregular characters (X, # and *) in V1 regions, from which 323 clade A, 2322 clade B, 1577 clade C, 618 CRF01_AE and 56 CRF07_BC Env sequences were isolated. The number of potential N-glycosylation sites (PNGS) in each extracted V1 region was predicted by N-GlycoSite program[65], and the V1 loop net charge was calculated with Bio.SeqUtils.IsoelectricPoint module in Biopython. A Python script was then devised to parse the V1 loop features (length, PNGS and net charge) into a dataframe (see Source Data Fig. 1g–i) to facilitate further analysis. By-subtype statistical summaries of V1 lengths and PNGS numbers were represented as box–whisker plots, and $p$ values were given by Mann–Whitney test. Differences were considered significant at $p < 0.05$, very significant at $p < 0.01$, highly significant at $p < 0.001$, and extremely significant at $p < 0.0001$. Box–whisker plots and pie charts were generated by GraphPad Prism 8.

### Correlation analysis between V1 features and bNAbs neutralization sensitivities

Neutralization data ($IC_{50}$ values reported as geometric means) of each bNAb and corresponding HIV-1 Env sequences were download from the Los Alamos HIV Database using CATNAP tool (http://hiv.lanl.gov/catnap). The V1 features (length and PNGS numbers) of downloaded sequences were extracted and prepared as described above and stored in Source Data Fig. 2c–g and Source Data Supplementary Fig. 3b. $Log^{IC50}$ was used for Kendall's tau correlation analysis with corresponding V1 length and PNGS. Data visualization was realized with the R package ggplot.

### F6 sensitivity signatures

To identify Env signatures associated with F6 sensitivity, published neutralization data of F6[20] (see Source Data Fig. 3e) and alignment of corresponding HIV-1 Env sequences (with HXB2 strain as reference sequence) were imported and analyzed with CATNAP: Custom Input[66]. Positional analysis was then performed for each F6-contacting Env residues to examine the correlation between F6 sensitivity and amino acid (AA) usage (and glycan presence) at this position. A Fisher's exact test is used to assess statistical significance and $p < 0.05$ was considered statistically significant. AA usage at each F6-contacting Env residues were displayed in WebLogo plots[67], wherein letter height represents AA frequencies. AAs associated with sensitivity and resistance to F6 are identified by Fisher's test and colored blue and red, respectively.

### Package of HIV-1 pseudoviruses (PsVs)

PsVs bearing X18 or 769 Env proteins were generated by co-transfecting HEK293T with plasmid pcDNA3.1-Env (encoding the C-terminal truncated Env proteins of X18 or 769 virus, oligos used for plasmids construction are summarized in Supplementary Table 5) and HIV backbone plasmid pNL4.3-HIV-luc at a mass ratio of 1:4. After overnight incubation, the transfected cells were incubated in fresh DMEM medium. Forty-eight hours after transfection, the supernatant was harvested by centrifugation at $1000 \times g$ for 10 min. Clarified supernatant was then collected, filtered, concentrated, and stored at −80 °C until further use.

### Analysis of pseudoviruses entry

Antibodies were first serially diluted and co-incubated with HIV-1 PsV for 1 h. Next, the virus and antibody mixtures were added into TZM-bl cells ($CD4^+CXCR4^+CCR5^+$) in 96-well plate. After 48 h, challenged cells were washed with PBS, lysed, and analyzed for intracellular luciferase activity with the Bright-Lite™ Luciferase Assay System (Vazyme, DD1204) according to manufacturer's instructions. Luminescence data was recorded with Tecan-Spark (Tecan) and analyzed with GraphPad Prism 9. Extended incubation neutralization assays were performed as described above but with extended incubation times for the virus and antibody mixtures.

### gp120 shedding assay

PsVs bearing X18 or Q769 Env proteins were incubated alone, or with 80 μg/ml sCD4 (or indicated Fabs) for different time periods at room temperature. Next, virions were pelleted by centrifugation at $22,000 \times g$ for 45 min. gp120 in the supernatant or remaining associated with the pelleted virions was analyzed with a gp120 capture enzyme-linked immunosorbent assay (ELISA). Briefly, samples were captured on 96-well plates pre-coated with capture antibody from Human Immunodeficiency Virus type 1 (HIV-1) gp120/Glycoprotein 120 ELISA Kit (Sino Biological, KIT11233, 1:1000) or Human Immunodeficiency Virus type 1 (HIV-1) gp120/Glycoprotein 120 Antibody, Rabbit PAb (Sino Biological, 11233-RP02, 1:1000) and detected using detection antibody from Human Immunodeficiency Virus type 1 (HIV-1) gp120/Glycoprotein 120

ELISA Kit (Sino Biological, KIT11233, 1:1000) or anti-gp120/gp160 (pan) (Immune Tech, IT-001-002M18, 1:1000). The percentage of gp120 remaining associated with virions was calculated with the equation:

$$\frac{\text{amount of gp120 associated with virions}}{\text{amount of gp120 in supernatant} + \text{amount of gp120 associated with virions}} \tag{1}$$

and normalized against untreated sample (100%) incubated for the same time.

### Reporting summary

Further information on research design is available in the Nature Portfolio Reporting Summary linked to this article.

## Data availability

The cryoEM maps have been deposited in the Electron Microscopy Data Bank with accession numbers: EMD-34190, EMD-34192, EMD-34193, and EMD-34194. The coordinates of F6 Fab, X18 UFO-8ANC195 trimer, X16 UFO-8ANC195 trimer, X18 UFO-F6 trimer (built upon the 4.14 Å cryo-EM map) and X18 UFO protomer-F6 (built upon the 4.05 Å map after local refinement) have been deposited to the Protein Data Bank (PDB) under accession numbers 8GPK, 8GPI, 8GPJ, 8GPG and 8GP5 respectively. All other data needed to evaluate the conclusions in the paper are present in the paper and/or the Supplementary Materials. Additional data related to this paper may be requested from the authors. Source data for Figs. 1e, 1g–i, 2c–g, 3e, 4d and Supplementary Figs. 1b, c, 3b, 4a are provided with this paper. Previously published structures used in this study are available in the PDB with accession numbers: 5BMF, 5CJX, 5FYK, 5UM8, 5FYJ, 5FUU, 4TVP, 6P65, 5CCK. Previously published CRF01_AE and CRF07_BC genome sequences mentioned in this study are available in the Genbank with accession numbers: AIS42911.1 and AIS42666.1. Source data are provided with this paper.

## Code availability

The Python and R scripts used in this study have been deposited to the Github repository at https://github.com/newjhon2013/HIV-project (https://doi.org/10.5281/zenodo.8135889)[68] with no restrictions for use.

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

## Acknowledgements

This work was supported by grants 2018ZX10731-101-001-010 (B.Y.), 2018ZX10731-101 (Y.M.S.), 2019YFA0802804 (B.Y.) and 2016YFE0107600 (Y.M.S.) from MoST, 32070170 (B.Y.) from National Natural Science Foundation of China (NSFC). This work was also supported in part by a Shanghai Municipal Education Commission (SMEC) grant to the Shanghai Frontiers Science Center for Biomacromolecules and Precision Medicine at the ShanghaiTech University. The authors thank the Discovery Technology Platform and the Bio-Electron Microscopy Facility at Shanghai Institute for Advanced Immunochemical Studies, ShanghaiTech University for instruments and technical support regarding SPR and EM experiments. The authors also thank the HPC Platform at ShanghaiTech University for high performance computing resources and the staffs of beamline BL19U1 at the National Center for Protein Science in Shanghai (NCPSS) for assistance during X-ray data collection. The X18 and X16 UFO constructs were designed and gifted by Prof. Jiang Zhu at the Scripps Research Institute.

## Author contributions

B.Y. and Y.M.S. conceived, designed, and supervised the project. J.N., Q.W., W.W.Z, B.M. and Y.W.X. performed most experiments with the help of X.F.Z., Y.F., Q.L.Q., Y.L.H., X.Z., Y.L., and J.C.X. on plasmids construction, cell culture and transfection, protein expression and purification. W.W.Z. packaged the PsVs and performed the shedding analysis and neutralization experiments, B.M. established the expression and purification protocol for X18 UFO and X16 UFO, Y.W.X. contributed in the cryo-EM condition optimization and cryo-EM data processing, J.N. and Q.W. performed all the other experiments. B.Y. and J.N. wrote the paper with inputs from the other authors. B.Y. managed the project.

## Competing interests

The authors declare no competing interests.
