## [Peer Review File · Nature Communications]

REVIEWER COMMENTS

Reviewer #1 (Remarks to the Author):

In “Structures and immune recognition of Env trimers from two Asia prevalent HIV-1 CRFs”, Niu et al. use uncleaved prefusion-optimized (UFO) design to make prefusion-closed trimers of CRF01_AE (X18 UFO) and CRF07_BC (X16 UFO) subtypes, which were complexed with the antigen-binding fragment (Fab) of antibody 8ANC195 and subject to cryo-EM structural analysis. Pairwise structural comparison revealed high similarity between X18 and X16 complexes and prior determined structures from clades A, B, C and G. The authors also examine strain-specific glycosylation features, confirming by mass spectrometry 30 of 30 glycan sites on X18 and 24 of 29 glycan sites on X16 and examining local shifts of strain-specific glycans. They focus on the V1 loop of X18, which they show is longer and has more N-linked glycosylated than V1s of clades A, B, and C, and further, that the V1 length and number of V1 PNGSs correlates inversely with neutralization potency of V3-bNAb PGT121, PGT128 and 10-1074 and V2-bNAb PG9, VRC26.25, and PGDM1400.

The authors then determine the structure of X18 in complex with antibody F6, which they show recognizes a gp120-gp41 interface epitope that includes glycan N88, and make much over a reduction in ordered density at the trimer apex, proposing trimer destabilization as an alternative mechanism of gp120-gp41 interface bNAb neutralization.

The new structures of Env trimers X18 and X16 with 8ANC195 as well as the new structure of X18 in complex with F6 (and the crystal structures of F6) are helpful additions to the field.

It is unclear, however, if F6 binding is impacting the apex (as suggested by the authors) or if F6 merely does not constrain the apex, and a disordered apex is the default for X18 (with 8ANC195 inducing a more stabilized apex). One way to distinguish between these two alternatives is to test the impact of the binding of F6 on the affinities of V2-apex binders. If F6 is truly disassembling the trimer apex, then V2-apex bNAb such as PG9, VRC26.25, or PGT145 should recognize the F6-bound Env with weaker affinity than to the unbound Env. In addition, it is unclear whether destabilization of the Env trimer apex leads to neutralization, as our current understanding of CD4-activation indicates that CD4 also destabilizes the trimer apex, as part of the standard entry process.

Other revisions the authors should make:

Lines 144-154. The authors compare the fusion peptide conformation of X18 and X16 with that of CRF01_AE T/F100, concluding that the conformations are different. However, the UFO design links gp120 to gp41, with a 2x(G4S) linker. Since the gp41 fusion peptide is right at the N-terminus of gp41, its conformation is influenced by both linker and lack of gp41 cleavage. The authors cannot thus conclude that the sequestered N terminus is unique to CRF01_AE T/F100, versus being altered by either linker or by lack of cleavage.

Lines 313-314. In addition to comparing F6 recognition to interface antibodies 8ANC195 and 35O22, the authors should compare to all published interface antibody-Env trimer complexes, including the very broad interface antibody induced by vaccination (1C2) (see Pauthner et al., 2019 Immunity).

Fig. 2c. Antibodies 10-1074 and PGT121 are from the same donor (and likely from the same lineages); it would be better to use a different V3-directed bNab to show the generality of the result.

Reviewer #2 (Remarks to the Author):

The study by Niu, Meng, Xu et al is an in depth, solid structural analysis of two recombinant, engineered HIV-1 Env trimers from two cross-clade recombinant CRF01_AE and CRF07_BC isolates. As the authors note, much of the available structural information available for HIV Env is based on clade B and C isolates, and for these even only a small few have been characterized. Expanding our understanding of HIV Env structural diversity is an important area because this begins to address the issue that has made HIV such a challenging target for vaccine development and for the use of nAb as therapeutics. Likewise, it is of interest to determine whether similar or different broadly neutralizing antibody responses are observed when the HIV=1 isolate is from one clade or another.

Thus in this work, one Env from a CRF01_AE and one from a CRF07_BC isolate are produced as engineered recombinant ectodomain trimers, and their structures are determined by cryo-EM with a previously described bnAb bound (8ANC195). In addition a structural analysis of an antibody, F6, isolated from a CRF01_AE patient case is examined in complex with the X18 CRF01_AE Env trimer. I appreciate the complementary mutagenesis work that was provided to further examine the contribution of various residues involved in the F6 epitope. Interestingly F6 is shown to induce the disassembly of the recombinant X18 trimer, suggesting a potential mechanism of neutralization. One other aspect of the study that is novel is an analysis of glycosylation abundance and V1 loop length for

Envs from the database in CRF01_AE and CRF07_BC clades, vs other subtypes. This is correlated against publically available neutralization sensitivity data (IC50 values). A moderate correlation was observed particularly for V3 antibodies, but I am not fully convinced that the other correlations described are significant as the spread in values is so large versus the trends observed.

The authors are somewhat handicapped in attempting to draw comparisons between their new structures and published structures because in addition to the Env sequence differences, in some cases the engineered changes are not the same (e.g. SOSIP vs NFL vs UFO, which include differences in gp120-gp41 cleavage, fusion peptide behavior, etc). They highlight some of the differences in HR1, which stem from differences in engineered changes in this region that are meant to bias conformation to the prefusion, closed state. The differences in fusion peptide are also difficult to draw much from since the NFL and UFO trimers are single chain (i.e. uncleaved between gp120 and gp41) rather than cleaved like the SOSIP cases.

Likewise, comparisons of F6 with other antibodies against the gp120-gp41 interfacial region don't all involve the same antigen target, thus even if differences are seen, it is not clear if it is due to the antibody or the antigen. (why not use the present X18 and X16 structural comparisons with 8ANC195 vs F6 for example? Instead an previous structure with 5CJX was used).

Thus, while there is a significant body of solid data provided by this study, I am not certain how far it necessarily conceptually advances our understanding of the diversity of HIV Env and antibody responses. Though the data provided are valuable additional new datapoints in a very large space that it is important to sample.

Specific points:

1) I would recommend the authors test for F6-induced gp120 shedding from cell surface Env or better, Env on virus, to validate mechanism of neutralization. Also would be useful to test against more than just the CRF01_AE Env

2) I may have missed this, but how do the contact residues for X16 and X18 with 8ANC195 Fab compare?

This is the type of comparison I think starts to get at interesting questions about broadly neutralizing antibodies. How do they recognize different antigenic targets?

Reviewer #3 (Remarks to the Author):

Niu and colleagues describe structural analysis of two Envs from CRF genotypes (for which structural information was previously lacking), based on cryo-EM structures and mass spec analysis presented in this manuscript. Unremarkably, the stabilized structures (UFO) agree well with previously solved structures for other Env genotypes. The major differences are inferred from sequence alignments (e.g. longer V1, more glycosylated). The work is well done but not ground breaking. Of higher interest is the antibody F6 which the authors characterize and note that it is the first isolated from an individual infected with a CRF. While the antibody is notable for causing trimer degradation, the epitope is not necessarily new nor is such behavior (dissociation) unknown in the field of HIV. The authors do make mention and some comparisons to similar antibodies, 3BC315 and 1C2 although most of the main text and figure comparisons are to the less-overlapping 8ANC195 or 35O22 (which does not have this property of destabilization).

Overall, I see no major flaws or insufficiencies in the research and conclusions. The cryo-EM work and findings are supported by the author-provided statistics and PDB validation reports. The manuscript can benefit from a more direct comparison to 1C2 and 3BC315, which will show that F6 is not the first antibody to this site of vulnerability:

1) Comparison of F6 to the cryoEM map (9 Angstrom) of 3BC315 in complex with BG505 SOSIP (EMD-3067). While a higher resolution map/model does not exist, its similarity to 1C2 (and being human-derived like F6) would make for a useful epitope comparison. Perhaps by fitting the 3BC315 crystal structure in the cryoEM map and comparing to the F6 coordinates in this manuscript.

2) While the KD of F6 is not affected by mutating out glycan N88 (N88A), the SPR data does reveal two important findings: the on-rate improves about 10-fold, while the off-rate speeds up 10-fold. This is very reminiscent to a similar finding for 3BC315 (PMID: 26404402; Jeong Hyun Lee et al Nat Comm 2015). "3BC176/3BC315, on the other hand, have slower on- and off-rates in the presence of the glycan. [N88]" Those authors proposed that N88 might act as a "clasp" that helps keep the antibody in place once bound. I encourage the authors to include the parallels to this study and speculate whether a similar behavior is occurring with F6.

3) It's worth mentioning that antibody-induced trimer dissociation also appears to be a relatively common observation in HIV vaccine studies, particularly with subunit (recombinant Env) immunogens (PMID: 34321200; Turner HL Sci Adv 2021).

4) For completeness, it may be worth comparing the epitope of F6 with PGT151, which is also a gp120/gp41 interface antibody.

5) The comparison to 1C2 (shown in a Extended Data 6) should be mentioned in the Results instead of briefly in Discussion.

Minor points:

Line 272: electron density is technically incorrect for a cryo-EM map, as it is a measure of Coulombic potential. Okay to simply say "the density of the Env apex..."

We thank all the reviewers for their thoughtful and insightful comments that have helped to improve this paper. We have addressed all points raised by the reviewers and our point-by-point responses are detailed below.

Reviewer #1 (Remarks to the Author):

In “Structures and immune recognition of Env trimers from two Asia prevalent HIV-1 CRFs”, Niu et al. use uncleaved prefusion-optimized (UFO) design to make prefusion-closed trimers of CRF01_AE (X18 UFO) and CRF07_BC (X16 UFO) subtypes, which were complexed with the antigen-binding fragment (Fab) of antibody 8ANC195 and subject to cryo-EM structural analysis. Pairwise structural comparison revealed high similarity between X18 and X16 complexes and prior determined structures from clades A, B, C and G. The authors also examine strain-specific glycosylation features, confirming by mass spectrometry 30 of 30 glycan sites on X18 and 24 of 29 glycan sites on X16 and examining local shifts of strain-specific glycans. They focus on the V1 loop of X18, which they show is longer and has more N-linked glycosylated than V1s of clades A, B, and C, and further, that the V1 length and number of V1 PNGSs correlates inversely with neutralization potency of V3-bNAbs PGT121, PGT128 and 10-1074 and V2-bNAbs PG9, VRC26.25, and PGDM1400. The authors then determine the structure of X18 in complex with antibody F6, which they show recognizes a gp120-gp41 interface epitope that includes glycan N88, and make much over a reduction in ordered density at the trimer apex, proposing trimer destabilization as an alternative mechanism of gp120-gp41 interface bNAbs neutralization.

The new structures of Env trimers X18 and X16 with 8ANC195 as well as the new structure of X18 in complex with F6 (and the crystal structures of F6) are helpful additions to the field.

Thanks very much for the overall supportive comments.

It is unclear, however, if F6 binding is impacting the apex (as suggested by the authors) or if F6 merely does not constrain the apex, and a disordered apex is the default for X18 (with 8ANC195 inducing a more stabilized apex). One way to distinguish between these two alternatives is to test the impact of the binding of F6 on the affinities of V2-apex binders. If

F6 is truly disassembling the trimer apex, then V2-apex bNAbs such as PG9, VRC26.25, or PGT145 should recognize the F6-bound Env with weaker affinity than to the unbound Env.

Thanks much for the helpful suggestion. In the revised manuscript, we have tested the impact of F6 binding on the affinities of V2-apex bNAbs with SPR experiments. As shown below, V2-apex bNAbs PGT145 and PG9 indeed recognize F6-bound Env trimers with 7~8 times lower affinities than to the unbound Env trimers. These results indicate that the binding of F6 is truly destabilizing and disassembling the trimer apex.

Binders	Envs	k_a ($M^{-1}s^{-1}$)	k_d (s^{-1})	K_D (M)	Relative affinity
PGT145	X18 UFO	2.23×10^4	8.17×10^{-4}	3.67×10^{-8}	100%
	X18 UFO + F6 (RT, 1-hr)	1.06×10^3	3.21×10^{-4}	3.02×10^{-7}	12.2%
PG9	X18 UFO	1.66×10^3	4.45×10^{-4}	2.67×10^{-7}	100%
	X18 UFO + F6 (RT, 1-hr)	3.64×10^2	6.36×10^{-4}	1.75×10^{-6}	15.3%

These new data have been incorporated into revised Supplementary Fig. 7a-7b, and discussed accordingly in the revised manuscript.

In addition, it is unclear whether destabilization of the Env trimer apex leads to neutralization, as our current understanding of CD4-activation indicates that CD4 also destabilizes the trimer apex, as part of the standard entry process.

Thanks much for this comment. Inspired by this comment, we first compared the conformational changes induced by CD4 (Gabriel Ozorowski *et al*, 2017 Cell) to that induced by F6 in apex regions. As shown below and in revised Supplementary Fig. 7, CD4 binding indeed induces ‘organized’ conformational rearrangement of the trimer apex, wherein the V1/V2 regions (red) and $\beta 20/\beta 21$ hairpin (green) of Env rearrange themselves to form the bridging sheet and liberate the V3 region (please see the left figure below). It is worth noting that, during this process, the V1/V2 stem region is not disordered and the interaction between it (red) and the $\beta 20/\beta 21$ hairpin (green) held them in correct positions, and together they constitute an important part of the CD4-contacting surface (five CD4-contacting Env segments indicated as red, green, grey, cyan and orange surfaces in the right figure below) on Env.

In contrast, F6-induced destabilization of the Env apex proximal regions appears much more extensive. As shown below and in revised Supplementary Fig. 7f, F6 binding destabilized the entire V1/V2 region and even part of the flanking C1/C2 regions (117-208 AA), including the V1/V2 stem. Moreover, the 421-439 AA of C4 region, which includes the entire $\beta 20/\beta 21$ hairpin, and part of the V5 loop (459-463 AA) also got destabilized upon F6 binding.

Of note, $\beta 20/\beta 21$ hairpin play key roles in CD4 engagement (Peter D. Kong *et al*, 1998 Nature; Gabriel Ozorowski *et al*, 2017 Cell). As shown below and in revised Supplementary Fig. 7d, W427 and V430 from the $\beta 20/\beta 21$ hairpin form hydrophobic interactions with F43^{CD4} and R59^{CD4}, two CD4 residues that are critical for Env binding (U Moebius *et al*, 1992 J Exp Med). Meanwhile, M426 from the $\beta 20/\beta 21$ hairpin H-bonds with S42^{CD4} through its main chain oxygen. Importantly, a single mutation at W427 in $\beta 20/\beta 21$ hairpin is sufficient to abrogate CD4 binding and render HIV-1 virus non-infectious (Agnes Cordonnier *et al*, 1989 Nature). Given such critical status of $\beta 20/\beta 21$ hairpin in CD4 engagement, it is

conceivable that F6-induced destabilization of the entire V1/V2 and $\beta 20/\beta 21$ hairpin regions would inevitably hinder the formation of bridging sheet (and thus the correct positioning of $\beta 20/\beta 21$ hairpin) and hamper the binding of CD4. Likewise, the F6 binding induced destabilization of 459-463AA (V5 region) would also damage the interaction between CD4 and Env in this region.

To test if F6-induced destabilization of Env apex proximal regions indeed hampers the binding of CD4, we compared the binding affinity between CD4 and X18 UFO to that between CD4 and F6-bound X18 UFO, and observed an over 80% affinity decrease when F6 is present (Please see revised Fig. 6a attached below). To make sure that this effect of F6 is rather general than strain- or subtype-specific, we also tested the impact of F6 on the binding of CD4 to Q769 Env trimer (clade A, tier 2), and a drastic decrease was observed when F6 is present (Please see revised Fig. 6a attached below). Together, these data indicate that F6-induced destabilization of trimer apex proximal regions would hinder the binding of receptor CD4, representing one potential neutralization mechanisms of F6.

Binders	Envs	k_a ($M^{-1}s^{-1}$)	k_d (s^{-1})	K_D (M)	Relative affinity
sCD4	X18 UFO	4.22×10^3	4.10×10^{-5}	9.72×10^{-9}	100%
	X18 UFO + F6 (RT, 1-hr)	1.56×10^3	9.30×10^{-5}	5.96×10^{-8}	16.3%
sCD4	Q769 UFO	5.82×10^4	7.01×10^{-5}	1.21×10^{-9}	100%
	Q769 UFO + F6 (RT, 1-hr)	1.92×10^4	2.64×10^{-4}	1.38×10^{-8}	8.8%

Destabilization of trimer apex proximal region is only the initial event triggered by F6. As time goes by, the F6-triggered protomers disengagement at both the apex and the base (please refer to revised Fig. 5a-5e for more details) would eventually lead to the disassembly of Env trimers (please see revised Fig. 5f and Supplementary Fig. 6c attached below).

The host entry of HIV-1 relies on successful host-viral membrane fusion. As type I fusion apparatus, Envs function as trimers, and the fusogenic 6-HB could not be formed properly if Env trimers disassemble into membrane-embedded monomers. Thus, when virus is incubated with disassembling antibody, its infectivity would gradually decrease as its Envs gets disassembled, and the apparent IC_{50} of the antibody would decrease correspondingly to incubation time (as more Envs would get disassembled and thus larger infectivity loss would be seen at longer time points). Consistent with such prediction, when virus is pre-incubated with F6, the IC_{50} of F6 for X18 and Q769 PsVs decreased 3.8- and 4.2-fold respectively over an 18-hr incubation time (please see revised Fig. 6b attached below), even greater than the decrease of the positive control sCD4 (2.6- and 3.4-fold, the IC_{50} decrease of sCD4 is due to sCD4-induced gp120 shedding). Meanwhile, the IC_{50} of 35O22, another interface antibody that does not cause gp120 shedding or trimer disassembly (Lee *et al.*, 2015 Nat Commun), showed little change during prolonged incubation. Collectively, these data suggest that the Env-disassembling ability of F6 likely represents the other neutralization mechanism of it.

X18 (CRF01_AE, Tier 2)						
Preincubation (hr)	IC ₅₀ (ug ml ⁻¹)	Fold change	IC ₅₀ (ug ml ⁻¹)	Fold change	IC ₅₀ (ug ml ⁻¹)	Fold change
	35O22		sCD4		F6	
1	0.021		3.87		40.87	
6	0.014	1.5	2.30	1.7	36.86	1.1
18	0.011	1.9	1.49	2.6	10.76	3.8

Q769 (Clade A, Tier 2)						
Preincubation (hr)	IC ₅₀ (ug ml ⁻¹)	Fold change	IC ₅₀ (ug ml ⁻¹)	Fold change	IC ₅₀ (ug ml ⁻¹)	Fold change
	35O22		sCD4		F6	
1	0.017		2.25		6.06	
6	0.016	1.1	1.04	2.2	2.82	2.2
18	0.011	1.5	0.66	3.4	1.45	4.2

We also summarized the proposed neutralization mechanism of F6 with a schematic diagram (please see revised Fig. 6d attached below).

Other revisions the authors should make:

Lines 144-154. The authors compare the fusion peptide conformation of X18 and X16 with that of CRF01_AE T/F100, concluding that the conformations are different. However, the UFO design links gp120 to gp41, with a 2x(G4S) linker. Since the gp41 fusion peptide is right at the N-terminus of gp41, its conformation is influenced by both linker and lack of gp41 cleavage. The authors cannot thus conclude that the sequestered N terminus is unique to CRF01_AE T/F100, versus being altered by either linker or by lack of cleavage.

Thanks for raising this point. We agree with the reviewer's comments and have rephrased corresponding sentences in the text as following:

“The FPs are generally ‘exposed and disordered’ in the closed state of prefusion Env trimers but would rearrange to become ‘buried’ in the CD4-induced open state of Env (Ozorowski, 2017). Consistent with their closed prefusion conformations, the FPs in X18 UFO and X16 UFO are both ‘exposed’, akin to the FP conformation in native prefusion Env trimers (Supplementary Fig. 1h, compared to JRFL WT). Previously, a rare ‘buried’ conformation of FP has been captured in a 3.9 Å prefusion structure of a very early transmitted founder virus (CRF01_AE T/F100), which was thought to represent an intermediate between the closed and open states (Ananthaswamy, 2019). Here, the FP of X18 UFO (also from CRF01_AE) did not recapitulate the ‘buried’ intermediate state seen in T/F100 SOSIP trimer, possibly due that the UFO trimer is not cleaved between gp120 and gp41 as in SOSIP trimer.”

Lines 313-314. In addition to comparing F6 recognition to interface antibodies 8ANC195 and 35O22, the authors should compare to all published interface antibody-Env trimer complexes, including the very broad interface antibody induced by vaccination (1C2) (see Pauthner et al., 2019 Immunity).

Thanks for the comment. Per reviewers' suggestions, we now compared the epitope of F6 to those of interface antibodies 35O22, 8ANC195, PGT151, 1C2 and 3BC315 in revised Fig. 4 (please also see the figures attached below). As shown below, the epitope of F6 is different from those defined by 35O22 and 8ANC195, two prototypical gp120-gp41 interface bNAbs, and is also distinct from the epitope of PGT151, which specifically sequesters the FPs of cleaved Env trimers. Meanwhile, the binding position of F6 on Env is similar to that of 3BC315, although their approaching angles and orientations of heavy and light chains are different. Intriguingly, the epitope of F6 overlaps by around 953 Å² to that of 1C2, a cross-reacting antibody (87% breadth) elicited by heterologous Env trimer-liposome prime:boosting in rabbits, and their approaching angles are very similar.

Different from 35O22, 8ANC195 and PGT151, wherein N-glycans make large and indispensable contributions to their epitopes, the epitopes of F6, 1C2 and 3BC315 only received limited and dispensable contributions from glycans (please see the right figure above). Indeed, the epitopes of F6, 1C2 and 3BC315 all locate near one of the two major functional gaps in the continuous glycan shield on Env surface. As discussed in our original submission, the fact that this glycan-free area is successfully captured by bNAbs from different clonal lineages and distinct HIV-1 infection background (3BC315: clade B, F6: CRF01_AE) demonstrates that the hosts can effectively appreciate this conserved vulnerability despite their diverse genetic background. Meanwhile, the fact that this area is also well captured by vaccination induced immunity in rabbits (*e.g.*, 1C2) further suggest that this area may be a feasible target site for immunofocusing vaccine design.

Although F6, 3BC315 and 1C2 recognize overlapping areas on Env surface, differences are obvious among them. As shown below and in revised Fig. 4b, in the context of Env trimer, 3BC315 and 1C2 interact with gp41 of two neighboring protomers simultaneously, while F6 only interacts with gp41 of one protomer. Besides, while 3BC315 binds Env trimer with maximum 2:1 stoichiometry (Lee *et al*, 2015, Nat Commun), 1C2 and F6 bind at 3:1 stoichiometry.

Moreover, as shown below and in revised Fig. 4c, F6 makes extensive and key interactions with gp120 residues, and single mutations at gp120 residues Y39, T499 and R500 lead to 97.3%, 90.5% and 90% affinity decrease of F6 respectively (please refer to revised Fig. 3d for more details). In contrast, rabbit antibody 1C2 only makes limited interactions with gp120, and bNAb 3BC315 barely interacts with gp120 residues (Lee *et al*, 2015, Nat Commun) (please see the revised Fig. 4c attached below, note that the CDRH3 of F6 inserts much deeper to reach gp120 residues than that of 3BC315 or 1C2).

The different participation of gp120 likely explains the differences in their neutralization mechanism, as 3BC315 induces gp120 shedding (Lee *et al*, 2015, Nat Commun) yet F6 does not (please see the revised Fig. 6c below), although they both can induce trimer disassembly.

Previously, V1/V2 apex, V3 or CD4bs bNAbs have all been found to frequently target highly

overlapping epitopes, although they were identified from different donors and belong to different clonal lineages (please refer to revised Supplementary Fig.9 for more information). In contrast, epitopes of gp120-gp41 interface bNAbs do not seem to converge except for bNAbs that recognize FP. Here, the identification of F6 epitope and its similarity to 3BC315 and 1C2 epitopes highlight that the glycan free area near their epitopes, akin to the CD4bs, V3 glycan supersite, V1/V2 apex and FP, is also a repeatedly targeted immune vulnerability on the Env surface.

Fig. 2c. Antibodies 10-1074 and PGT121 are from the same donor (and likely from the same lineages); it would be better to use a different V3-directed bNAb to show the generality of the result.

Thanks for the helpful suggestions. Indeed, antibodies 10-1074 and PGT121 are both from donor 7 and are of the same lineage. To address this issue, we have now used the neutralization data of a different V3-directed bNAb lineage, DH270 (donor CH848), to calculate the correlation. As shown below, akin to prototypical V3 bNAbs PGT128 (donor 36, lineage PGT128) and PGT121/10-1074 (donor 7, lineage PGT121), the neutralizing potency of DH270 (donor CH848, lineage DH270) bNAbs manifested inverse correlations to V1 length ($p=1.8 \times 10^{-6}$). Also, more PNGS in V1 correlates with decreased sensitivity to V3 bNAbs PGT128, 10.1074 and DH270 ($p=5.8 \times 10^{-5}$), though not to PGT121.

We have incorporated the new data of DH270 into revised Fig. 2c and moved the data of PGT121 into Supplementary Fig. 3b. Related figures legends and discussions have also been modified accordingly in the revised manuscript.

Reviewer #2 (Remarks to the Author):

The study by Niu, Meng, Xu et al is an in depth, solid structural analysis of two recombinant, engineered HIV-1 Env trimers from two cross-clade recombinant CRF01_AE and CRF07_BC isolates. As the authors note, much of the available structural information available for HIV Env is based on clade B and C isolates, and for these even only a small few have been characterized. Expanding our understanding of HIV Env structural diversity is an important area because this begins to address the issue that has made HIV such a challenging target for vaccine development and for the use of nAb as therapeutics. Likewise, it is of interest to determine whether similar or different broadly neutralizing antibody responses are observed when the HIV-1 isolate is from one clade or another.

Thus in this work, one Env from a CRF01_AE and one from a CRF07_BC isolate are produced as engineered recombinant ectodomain trimers, and their structures are determined by cryo-EM with a previously described bnAb bound (8ANC195). In addition a structural analysis of an antibody, F6, isolated from a CRF01_AE patient case is examined in complex with the X18 CRF01_AE Env trimer. I appreciate the complementary mutagenesis work that was provided to further examine the contribution of various residues involved in the F6 epitope. Interestingly F6 is shown to induce the disassembly of the recombinant X18 trimer, suggesting a potential mechanism of neutralization. One other aspect of the study that is novel is an analysis of glycosylation abundance and V1 loop length for Envs from the database in CRF01_AE and CRF07_BC clades, vs other subtypes. This is correlated against publically available neutralization sensitivity data (IC₅₀ values). A moderate correlation was observed particularly for V3 antibodies, but I am not fully convinced that the other correlations described are significant as the spread in values is so large versus the trends observed.

Thanks very much for the comments of this reviewer.

The authors are somewhat handicapped in attempting to draw comparisons between their new structures and published structures because in addition to the Env sequence differences, in some cases the engineered changes are not the same (e.g. SOSIP vs NFL vs UFO, which include differences in gp120-gp41 cleavage, fusion peptide behavior, etc). They highlight

some of the differences in HR1, which stem from differences in engineered changes in this region that are meant to bias conformation to the prefusion, closed state. The differences in fusion peptide are also difficult to draw much from since the NFL and UFO trimers are single chain (i.e. uncleaved between gp120 and gp41) rather than cleaved like the SOSIP cases.

We agree with the reviewer's comment on HR1_N and fusion peptide and have rephrased corresponding sentences in the text as following:

“While the structural differences in the hypervariable loops are consistent with their high sequence variations, the conformational differences in the HR1_N region is largely attributable to the adoption of native-like trimer designs (Kumar, 2020), wherein the engineering of proximal regions in SOSIP (Sanders, 2013), NFL (Sharma, 2015) or UFO (Kong, 2016) design all disrupt the native α -helical conformation of HR1_N (Supplementary Fig. 1g, compare others to JRFL WT). The FPs are generally ‘exposed and disordered’ in the closed state of prefusion Env trimers but would rearrange to become ‘buried’ in the CD4-induced open state of Env (Ozorowski, 2017). Consistent with their closed prefusion conformations, the FPs in X18 UFO and X16 UFO are both ‘exposed’, akin to the FP conformation in native prefusion Env trimers (Supplementary Fig. 1h, compared to JRFL WT). Previously, a rare ‘buried’ conformation of FP has been captured in a 3.9 Å prefusion structure of a very early transmitted founder virus (CRF01_AE T/F100), which was thought to represent an intermediate between the closed and open states (Ananthaswamy, 2019). Here, the FP of X18 UFO (also from CRF01_AE) did not recapitulate the ‘buried’ intermediate state seen in T/F100 SOSIP trimer, possibly due that the UFO trimer is not cleaved between gp120 and gp41 as in SOSIP trimer.”

Likewise, comparisons of F6 with other antibodies against the gp120-gp41 interfacial region don't all involve the same antigen target, thus even if differences are seen, it is not clear if it is due to the antibody or the antigen. (why not use the present X18 and X16 structural comparisons with 8ANC195 vs F6 for example? Instead an previous structure with 5CJX was used).

Thanks for the comment. Theoretically, as a bNAb, 8ANC195 will bind different Envs in similar ways. Indeed, the interface between 8ANC195 and X18 UFO or X16 UFO is largely the same as that between 8ANC195 and BG505 Env (please refer to revised Supplementary Fig. 2 and our response to specific point 2 for more information). Per reviewer's suggestion, we have now used the structure of 8ANC195-X18 UFO from this study for interface antibody comparison instead in revised Fig. 4 (also attached below for reviewer's information), and our conclusion regarding F6 and 8ANC195 comparison remains unchanged.

Thus, while there is a significant body of solid data provided by this study, I am not certain how far it necessarily conceptually advances our understanding of the diversity of HIV Env and antibody responses. Though the data provided are valuable additional new datapoints in a very large space that it is important to sample.

Thanks very much for the comments.

Specific points:

1) I would recommend the authors test for F6-induced gp120 shedding from cell surface Env or better, Env on virus, to validate mechanism of neutralization. Also would be useful to test against more than just the CRF01_AE Env.

Thanks for this helpful suggestion. In the revised manuscript, we performed gp120 shedding from pseudotyped X18 (CRF01_AE) and Q769 (clade A) viruses. As shown below and in revised Fig.6c, we found that F6 does not induce gp120 shedding.

Such result is consistent with the fact that F6 makes extensive and key interactions with gp120 (please refer to revised Fig. 3b, close-up views of F6-Env interface, for more information). Indeed, when single mutations are introduced to gp120 residues Y39, T499 or R500, 97.3%, 90.5% and 90% decreases in F6 binding affinity were observed respectively (please refer to revised Fig. 3d for more details) and gp120 residue 500 has even been identified to be a key signature for HIV-1 viruses' sensitivity to F6 (please refer to revised Fig. 3e-3g for more information). Given such crucial interactions between F6 and gp120, it is understandable that F6 does not induce gp120 shedding.

To further reveal the detailed working mechanism of F6-mediated neutralization, we then performed the following experiments. In the original manuscript, we have shown that the binding of F6 renders the trimer apex proximal regions destabilized (please refer to Fig. 3a and Fig. 5 in the revised manuscript for more information). In the revised manuscript, we further compared the affinities of V2-apex bNAbs to Env in the absence and presence of F6 to prove that F6 binding is truly destabilizing the trimer apex. As shown below, V2-apex bNAbs PGT145 and PG9 recognize F6-bound X18 UFO with substantially lower affinities than to the unbound Env.

Binders	Envs	k_a ($M^{-1}s^{-1}$)	k_d (s^{-1})	K_D (M)	Relative affinity
PGT145	X18 UFO	2.23×10^4	8.17×10^{-4}	3.67×10^{-8}	100%
	X18 UFO + F6 (RT, 1-hr)	1.06×10^3	3.21×10^{-4}	3.02×10^{-7}	12.2%
PG9	X18 UFO	1.66×10^3	4.45×10^{-4}	2.67×10^{-7}	100%
	X18 UFO + F6 (RT, 1-hr)	3.64×10^2	6.36×10^{-4}	1.75×10^{-6}	15.3%

Such observation not only confirms that the binding of F6 is truly destabilizing the trimer apex, but also indicates that destabilization of corresponding regions would hinder the binding of

proteins that recognize these regions. We thus went on to analyze if F6-destabilized regions include regions involved in Env function. Indeed, F6 induced extensive destabilization of the Env apex proximal regions. As shown below and in revised Supplementary Fig. 7f, F6 binding destabilized the entire V1/V2 region and even part of the flanking C1/C2 regions (117-208 AA), including the V1/V2 stem. Moreover, the 421-439 AA of C4 region, which includes the entire $\beta 20/\beta 21$ hairpin, and part of the V5 loop (459-463 AA) also got destabilized upon F6 Binding.

Notably, V1/V2 stem, $\beta 20/\beta 21$ hairpin and the V5 region all play key roles in CD4 engagement (Peter D. Kong *et al*, 1998 Nature; Gabriel Ozorowski *et al*, 2017 Cell). As shown below and in revised Supplementary Fig. 7c-7d, CD4 (yellow ribbons) is embraced by a depression formed by five segments of gp120: $\beta 2/\beta 3$ (V1/V2 stem, red), loop D (C2, cyan), $\alpha 3/\beta 15$ excursion (C3, grey), $\beta 20/\beta 21$ hairpin (C4, green) and $\beta 23/V5/\beta 24$ (C4/V5/C5, orange). Of the five segments, $\beta 20/\beta 21$ hairpin appears very important. As shown in the right figure below, W427 and V430 from the $\beta 20/\beta 21$ hairpin form hydrophobic interactions with F43^{CD4} and R59^{CD4}, two CD4 residues that are critical for Env binding (U Moebius *et al*, 1992 J Exp Med). Meanwhile, M426 from the $\beta 20/\beta 21$ hairpin H-bonds with S42^{CD4} through its main chain oxygen. More importantly, a single mutation at W427 ^{$\beta 20/\beta 21$} in Env is sufficient to abrogate CD4 binding and render HIV-1 virus non-infectious (Agnes Cordonnier *et al*, 1989 Nature).

Given such critical status of $\beta 20/\beta 21$ hairpin in CD4 engagement, it is conceivable that F6-induced destabilization of the V1/V2 and $\beta 20/\beta 21$ hairpin regions would inevitably hamper the binding of CD4. Likewise, the F6 binding induced destabilization of 459-463AA (V5 region) would also damage the interaction between Env and CD4 in this region. Indeed, the binding of F6 to Env lead to drastic affinity decrease of CD4 (please see revised Fig. 6a attached below).

Binders	Envs	k_a ($M^{-1}s^{-1}$)	k_d (s^{-1})	K_D (M)	Relative affinity
sCD4	X18 UFO	4.22×10^3	4.10×10^{-5}	9.72×10^{-9}	100%
	X18 UFO + F6 (RT, 1-hr)	1.56×10^3	9.30×10^{-5}	5.96×10^{-8}	16.3%
sCD4	Q769 UFO	5.82×10^4	7.01×10^{-5}	1.21×10^{-9}	100%
	Q769 UFO + F6 (RT, 1-hr)	1.92×10^4	2.64×10^{-4}	1.38×10^{-8}	8.8%

Together, these data indicate that F6-induced destabilization of apex proximal regions would hinder the binding of receptor CD4, representing one potential neutralization mechanisms of F6.

Nevertheless, destabilization of apex proximal region is only the initial event triggered by F6. As time goes by, the F6 binding induced protomers disengagement at both the apex and the base (please refer to revised Fig. 5a-e for more information) would eventually lead to the disassembly of Env trimers (please see below and also revised Fig. 5f, Supplementary Fig. 6c).

The host entry of HIV-1 relies on successful host-viral membrane fusion. As type I fusion apparatus, Envs function as trimers, and the fusogenic 6-HB could not be formed properly if Env trimers disassemble into membrane-embedded monomers. Thus, when virus is incubated with disassembling antibody, its infectivity would gradually decrease as its Env gets disassembled, and the apparent IC_{50} of the antibody would decrease correspondingly to incubation time (as more Envs would get disassembled and thus larger infectivity loss would be seen at longer time points) (Lee *et al.*, 2015, Nat Commun). Consistent with such prediction, when virus is pre-incubated with F6, the IC_{50} of F6 for X18 and Q769 PsVs decreased 3.8- and 4.2-fold respectively over an 18-hr incubation time (please see revised Fig.6b attached below), even greater than the decrease of the positive control sCD4 (2.6- and 3.4-fold, the IC_{50} decrease of sCD4 is rendered by CD4-induced gp120 shedding). Meanwhile, the IC_{50} of 35O22, another interface antibody that does not cause gp120 shedding or trimer disassembly (Lee *et al.*, 2015, Nat Commun), showed little change during prolonged incubation.

X18 (CRF01_AE, Tier 2)						
Preincubation (hr)	IC ₅₀ (ug ml ⁻¹)	Fold change	IC ₅₀ (ug ml ⁻¹)	Fold change	IC ₅₀ (ug ml ⁻¹)	Fold change
	35O22		sCD4		F6	
1	0.021		3.87		40.87	
6	0.014	1.5	2.30	1.7	36.86	1.1
18	0.011	1.9	1.49	2.6	10.76	3.8

Q769 (Clade A, Tier 2)						
Preincubation (hr)	IC ₅₀ (ug ml ⁻¹)	Fold change	IC ₅₀ (ug ml ⁻¹)	Fold change	IC ₅₀ (ug ml ⁻¹)	Fold change
	35O22		sCD4		F6	
1	0.017		2.25		6.06	
6	0.016	1.1	1.04	2.2	2.82	2.2
18	0.011	1.5	0.66	3.4	1.45	4.2

Collectively, these data suggest that the Env-disassembling ability of F6 likely represents an alternative neutralization mechanism of it. We also summarized the proposed neutralization mechanism of F6 with a schematic diagram (please see revised Fig. 6d attached below).

2) I may have missed this, but how do the contact residues for X16 and X18 with 8ANC195 Fab compare? This is the type of comparison I think starts to get at interesting questions about broadly neutralizing antibodies. How do they recognize different antigenic targets?

Thanks for this question. To address this point, we compared the interfaces between 8ANC195 and X18 UFO (CRF01_AE), X16 UFO (CRF07_BC) and BG505 Env (subtype A) in the revised Supplementary Fig. 2. In all three cases, the conformation and position of 8ANC195-contacting Env segments (shown as ribbons on surface presentations of HIV-1 Env protomer, BG505:grey, X18 UFO:cyan, X16 UFO:wheat) remain unchanged (please see the Supplementary Fig. 2a attached below), despite a few amino acids variations in these Env segments (shown as sticks and labeled).

In general, these amino acid variations of interface residues among the three Envs do not affect the binding of 8ANC195 much. For instance, the replacement of I277 in BG505 to L277 in X18 and X16 would not change the hydrophobic interactions between the FR3 of 8ANC195 and Env segments 275-280 aa and 351-354 aa (please see the revised Supplementary Fig. 2c attached below). Likewise, the replacement of E92, P240, and R617 in BG505 Env by N92, K240 or H240, and K617 in X18 UFO and X16 UFO would not compromise much the electrostatic interactions (e.g., between the side chains of D92^{CDRL3} and K617^{Env} or R617^{Env}) and hydrophobic interactions (e.g., between the aliphatic part of R54^{CDRH2} side chain and the side chains of P240^{Env}, K240^{Env} or H240^{Env}) between indicated regions (please see the revised Supplementary Fig. 2d-2e attached below).

Similarly, as shown in revised Supplementary Fig. 2f (also attached below for reviewer's information), the replacement of K46, D47, L629, Q630 in BG505 Env (left) by R46, E47, I629/M629, E630 in X18 UFO (middle) and X16 UFO (right) does not compromise their interactions with W100^A, Y98, K100 and D100^G from 8ANC195 respectively. Meanwhile, although the replacement of K633 in BG505 Env (left) by E633 in X16 UFO (right, labeled in red to highlight) breaks its original packing with H100^E (CDRH3) and W32 (CDRL1), it enables new interaction with R50 (CDRL2) from 8ANC195 instead.

Hence, 8ANC195 binds X18 (CRF01_AE) or X16 (CRF07_BC) in similar ways and strength as it binds BG505 (clade A) Env, corroborating its role as a bNAb. To find out if in general CRF01_AE and CRF07_BC viruses would manifest altered sensitivities to 8ANC195, we also calculated the amino acid (AA) usage frequencies of CRF01_AE (n=618) and CRF07_BC (n=56) viruses in 8ANC195-contacting region and compared them to Env signatures associated with 8ANC195 sensitivity (please see revised Supplementary Fig. 2g attached below).

As shown above, most of the 8ANC195-contacting Env residues are conserved among different viruses. Meanwhile, for those residues that do show amino acids variations across different viruses, CRF01_AE and CRF07_BC viruses did not exhibit particular enrichment of 8ANC195 sensitive (blue) or resistant (red) AA signatures as compared to other M-group viruses, indicating that the sensitivities of CRF01_AE and CRF07_BC viruses to 8ANC195 would not significantly differ from other clades.

Reviewer #3 (Remarks to the Author):

Niu and colleagues describe structural analysis of two Envs from CRF genotypes (for which structural information was previously lacking), based on cryo-EM structures and mass spec analysis presented in this manuscript. Unremarkably, the stabilized structures (UFO) agree well with previously solved structures for other Env genotypes. The major differences are inferred from sequence alignments (e.g. longer V1, more glycosylated). The work is well done but not ground breaking. Of higher interest is the antibody F6 which the authors characterize and note that it is the first isolated from an individual infected with a CRF. While the antibody is notable for causing trimer degradation, the epitope is not necessarily new nor is such behavior (dissociation) unknown in the field of HIV. The authors do make mention and some comparisons to similar antibodies, 3BC315 and 1C2 although most of the main text and figure comparisons are to the less-overlapping 8ANC195 or 35O22 (which does not have this property of destabilization).

Overall, I see no major flaws or insufficiencies in the research and conclusions. The cryo-EM work and findings are supported by the author-provided statistics and PDB validation reports. The manuscript can benefit from a more direct comparison to 1C2 and 3BC315, which will show that F6 is not the first antibody to this site of vulnerability:

Thanks very much for the comments of this reviewer.

1) Comparison of F6 to the cryoEM map (9 Angstrom) of 3BC315 in complex with BG505 SOSIP (EMD-3067). While a higher resolution map/model does not exist, its similarity to 1C2 (and being human-derived like F6) would make for a useful epitope comparison. Perhaps by fitting the 3BC315 crystal structure in the cryoEM map and comparing to the F6 coordinates in this manuscript.

Thanks for the helpful suggestion and detailed instructions. We have fitted the crystal structure of 3BC315 into the cryoEM map of it in complex with BG505 Env (EMD-3067) and used the resulting model to compare the epitopes of different interface antibodies, including F6, 8ANC195, 35O22, PGT151, 1C2 and 3BC315 in revised Fig. 4 (also attached below for reviewer's information). The similarities and differences of these antibodies have also been discussed in corresponding Results section.

As shown below, the epitope of F6 is distinct from those defined by 35O22, 8ANC195, and PGT151. Meanwhile, the binding position of F6 on Env is similar to that of 3BC315, although their approaching angles and orientations of heavy and light chains are different. Intriguingly, the epitope and approaching angle of F6 are very similar to that of 1C2, a cross-reacting antibody (87% breadth) elicited by heterologous Env trimer-liposome prime:boosting in rabbits.

Different from 35O22, 8ANC195 and PGT151, wherein N-glycans make large and indispensable contributions to their epitopes, the epitopes of F6, 3BC315 and 1C2 only received limited and dispensable contributions from glycans (please see the right figure above). Indeed, the epitopes of F6, 3BC315 and 1C2 all locate near one of the two major functional gaps in the continuous glycan shield on Env surface. As discussed in the original submission, the fact that this glycan-free area is successfully captured by bNAbs from different clonal lineages and distinct HIV-1 infection background (3BC315: clade B, F6: CRF01_AE) demonstrates that the hosts can effectively appreciate this conserved vulnerability despite their diverse genetic background. Meanwhile, the fact that this area is also well captured by vaccination induced immunity in rabbits (*e.g.*, 1C2) further suggest that this area may be a feasible target site for immunofocusing vaccine design.

Although F6, 3BC315 and 1C2 recognize overlapping areas on Env surface, differences are obvious among them. As shown below and in the revised Fig. 4b, in the context of Env trimer, 3BC315 and 1C2 interact with gp41 of two neighboring protomers simultaneously, while F6

only interacts with gp41 of one protomer. Besides, while 3BC315 binds Env trimer with maximum 2:1 stoichiometry (Lee *et al*, 2015, Nat Commun), 1C2 and F6 bind at 3:1 stoichiometry.

Moreover, as shown below and in revised Fig. 4c, F6 makes extensive and key interactions with gp120 residues, and single mutations at gp120 residues Y39, T499 and R500 lead to 97.3%, 90.5% and 90% affinity decrease of F6 respectively (please refer to revised Fig. 3d for more details). In contrast, rabbit antibody 1C2 only makes limited interactions with gp120, and bNAb 3BC315 barely interacts with gp120 residues (Lee *et al*, 2015, Nat Commun) (please see the revised Fig. 4c attached below, note that the CDRH3 of F6 inserts much deeper to reach gp120 residues than that of 3BC315 or 1C2).

The different participation of gp120 also explains the differences in their neutralization mechanism, as 3BC315 induces gp120 shedding (Lee *et al*, 2015 Nat Commun) yet F6 does not (please see the revised Fig. 6c attached below), although they both can induce trimer disassembly.

Previously, V1/V2 apex, V3 or CD4bs bNAbs have all been found to frequently target highly overlapping epitopes, although they were identified from different donors and belong to different clonal lineages (please refer to revised Supplementary Fig.9 for more information). In contrast, epitopes of gp120-gp41 interface bNAbs does not seem to converge except for bNAbs that recognize FP. Here, the identification of F6 epitope and its similarity to 3BC315 and 1C2 epitopes highlight that the glycan free area near their epitopes, akin to the CD4bs, V3 glycan supersite, V1/V2 apex and FP, is also a repeatedly targeted immunogenic vulnerability on the Env surface.

2) While the K_D of F6 is not affected by mutating out glycan N88 (N88A), the SPR data does reveal two important findings: the on-rate improves about 10-fold, while the off-rate speeds up 10-fold. This is very reminiscent to a similar finding for 3BC315 (PMID: 26404402; Jeong Hyun Lee et al Nat Comm 2015). “3BC176/3BC315, on the other hand, have slower on- and off-rates in the presence of the glycan. [N88]” Those authors proposed that N88 might act as a “clasp” that helps keep the antibody in place once bound. I encourage the authors to include the parallels to this study and speculate whether a similar behavior is occurring with F6.

Thanks for the suggestion. We agree with the reviewer’s comments and have rephrased corresponding sentences in the revised main text as following:

“When N88 is mutated to Ala, the association rate between F6 and X18 UFO roughly doubles, while the disassociation rate also speeds up about 2-fold, leaving the K_D largely unchanged (Fig. 3d and Supplementary Fig. 5a). Such phenomenon is reminiscent of 3BC315, a gp41-binding bNAb isolated from clade B infected individuals, which also shows synchronously

increased on- and off-rates in the absence of glycan (Lee, 2015). Hence, as previously proposed, although N88 glycan is not required for F6 and 3BC315 to bind and may even sterically restrict their access, it may work as a 'clasp' to help keep antibodies in place once they are bound (Lee, 2015)."

3) It's worth mentioning that antibody-induced trimer dissociation also appears to be a relatively common observation in HIV vaccine studies, particularly with subunit (recombinant Env) immunogens (PMID: 34321200; Turner HL Sci Adv 2021).

Thanks much for the suggestion. We now have discussed about the mentioned studies in the revised manuscript as following:

"Notably, induced disassembly of Env trimers has also been observed for 3BC315 and 1C2 (Lee, 2015; Dubrovskaya, 2019), as well as non- or weakly-neutralizing antibodies elicited by subunit immunogens in vaccine studies (Turner, 2021). The latter antibodies approach the Env at an angle that is incompatible with the membrane and thus would not be able to disassemble membrane-embed Envs on virions, explaining their lack of neutralizing activity (Turner, 2021)."

4) For completeness, it may be worth comparing the epitope of F6 with PGT151, which is also a gp120/gp41 interface antibody.

Thanks for the suggestion. We have now compared the epitope of F6 to that of interface antibodies 8ANC195, 35O22, PGT151, 1C2 and 3BC315 in the revised Fig. 4 and discussed their similarities and differences accordingly. Please also refer to our response to specific point 1 for more details.

5) The comparison to 1C2 (shown in a Extended Data 6) should be mentioned in the Results instead of briefly in Discussion.

Thanks for the comments. Per reviewers' suggestions, we now have compared the epitope of F6 to those of interface antibodies 8ANC195, 35O22, PGT151, 1C2 and 3BC315 in the revised Fig. 4 and discussed about their similarities and differences in corresponding Results section. Please also refer to our response to specific point 1 for more details.

Minor points:

Line 272: electron density is technically incorrect for a cryo-EM map, as it is a measure of Coulombic potential. Okay to simply say "the density of the Env apex..."

Thanks, fixed.

REVIEWERS' COMMENTS

Reviewer #1 (Remarks to the Author):

The authors have robustly responded to reviewer comments, and the revised manuscript is improved.

In addition to the new structures of Env and F6, the mechanism of N6, destabilizing the trimer apex by binding at the gp120-gp41 interface, is quite fascinating, showing how binding at more membrane-proximal portions of the trimer can impact the apex.

The authors state on lines 407-409 that: “Destabilization of apex proximal region is only the initial event triggered by F6. As time goes by, the disengagement of protomers at both the apex and the base would eventually lead to the disassembly of Env trimers (Fig. 5)...”. However, in Fig. 6, there is no indication that F6 induces gp120 dissociation. The authors should therefore temper their comments on line 407-409 in light of the absence of experimental confirmation for F6-induced gp120 shedding – and modify the Fig. 6d schematic to clarify that trimer disassembly does not include gp120 shedding.

My only remaining suggestion is to make sure Env numbering is in the standard Env numbering defined in the Los Alamos Database by Bette Korber to ease the comparison of these structures with others. Also, that F6 is in Kabat numbering to ease comparison with other antibodies.

Reviewer #2 (Remarks to the Author):

I appreciate the new experiments and analysis the authors have provided. The overall study is more complete now and addresses my main concerns from the original submission.

Reviewer #3 (Remarks to the Author):

The authors carefully addressed all of the points from my initial review, with updated figures and text that boost the quality of the manuscript. I congratulate the authors on their work and look forward to its publication to the HIV vaccine community.

We thank the reviewers for their supportive comments and helpful suggestions. We have addressed all the remaining concerns and our point-to-point responses are detailed below.

Reviewer #1 (Remarks to the Author):

The authors have robustly responded to reviewer comments, and the revised manuscript is improved.

In addition to the new structures of Env and F6, the mechanism of N6, destabilizing the trimer apex by binding at the gp120-gp41 interface, is quite fascinating, showing how binding at more membrane-proximal portions of the trimer can impact the apex.

Many thanks for the supportive comments.

The authors state on lines 407-409 that: “Destabilization of apex proximal region is only the initial event triggered by F6. As time goes by, the disengagement of protomers at both the apex and the base would eventually lead to the disassembly of Env trimers (Fig. 5)...”. However, in Fig. 6, there is no indication that F6 induces gp120 dissociation. The authors should therefore temper their comments on line 407-409 in light of the absence of experimental confirmation for F6-induced gp120 shedding – and modify the Fig. 6d schematic to clarify that trimer disassembly does not include gp120 shedding.

Thanks for the helpful suggestion. Consistent with reviewer’s comments, our experimental data in Fig. 6 indicate that F6 does not induce gp120 shedding. Instead, two events are triggered by F6 binding: 1) destabilization of Env apex proximal regions (the initial event) and 2) disassembly of Env trimers into gp120-gp41 protomers (the subsequent event). We realized that our initial comments on lines 407-409 and drawing in Fig.6d may cause misunderstanding. To provide further clarification, we thus revised lines 407-409 as following (with edits in red):

*“Destabilization of apex proximal region is only the initial event triggered by F6. As time goes by, the disengagement of **gp120-gp41** protomers at both the **trimer** apex and base would eventually lead to the disassembly of Env trimers **into gp120-gp41 protomers** (Fig. 5)...”.*

Meanwhile, to illustrate the working mechanism of F6 more clearly, we have modified Fig. 6d as below:

The legend of Fig.6d has also been revised as following (with edits in red):

“d Schematic diagram illustrating the dual neutralization mechanism of F6: 1) F6 binding first destabilizes the apex proximal regions of Env (F6-bound intermediate state), thereby hindering the binding of CD4 receptor; 2) F6-induced destabilization of trimer apex and gp120-gp41 protomers disengagement at the trimer base finally lead to the disassembly of Env trimer into gp120-gp41 protomers (F6-bound final state), disassembled Env trimer loses its ability to form fusogenic six-helical bundle (6HB) and thus cannot mediate efficient host-viral membrane fusion.”

We hope that the revised Fig.6d and explanations in lines 407-409 are less confusing and provide clearer explanation of F6’s working mechanism.

My only remaining suggestion is to make sure Env numbering is in the standard Env numbering defined in the Los Alamos Database by Bette Korber to ease the comparison of these structures with others. Also, that F6 is in Kabat numbering to ease comparison with other antibodies.

We sincerely appreciate the reviewer’s helpful comments in this matter. We confirm that the Env numbering is in the standard Env numbering defined in the Los Alamos Database by Bette Korber. Indeed, Env sequences used in this study were all aligned to the HXB2-K03455 Env sequence with the HIValign tool (<https://www.hiv.lanl.gov/content/sequence/VIRALIGN/viralalign.html>) and numbered accordingly following the instructions in "Numbering Positions in HIV Relative to HXB2CG" by Bette T. Korber *et al.* We also made sure that the F6 antibody is in Kabat numbering, and Fig. 4e wherein the CDRH3 sequences does not conform to Kabat numbering has been updated accordingly (also attached below for easy access).

	CDRH3 sequence	HC V gene	LC V gene	CDRH3 length	CDRL3 length	SHM (%)	infected with
F6	WRLMMVDEVTRHGMDV	4-34*01	κV2-28*01	16	9	14	CRF01_AE
35O22	GLLRDGSSTWLPYL	1-18*03	λV2-14*02	14	10	22	clade B
8ANC195	TSTYYDRWSGLHHGVMAPSS	1-3*03	κV1-5*03	20	9	21	clade B
PGT151	MFQESGPPRLDRWSGRNYYYYSGMDV	3-20*03	κV2D-2*029	26	9	16	clade C
3BC315	PMRPVSHGIDYSGLFVFQF	1-2	κV2-23	19	10	/	clade B

Reviewer #2 (Remarks to the Author):

I appreciate the new experiments and analysis the authors have provided. The overall study is more complete now and addresses my main concerns from the original submission.

Many thanks for the supportive comments.

Reviewer #3 (Remarks to the Author):

The authors carefully addressed all of the points from my initial review, with updated figures and text that boost the quality of the manuscript. I congratulate the authors on their work and look forward to its publication to the HIV vaccine community.

Many thanks for the supportive comments.